# AMSC: ADAPTIVE MULTI-DIMENSIONAL STRUCTURED COMPRESSION WITH THEORETICAL GUARANTEES

## ABSTRACT

Network pruning is a pivotal strategy for reducing complexity and accelerating inference. Most pruning methods focus on a single dimension (depth or width), leading to insufficient compression when multiple dimensions are redundant. Additionally, separating pruning from training disrupts established network correlations, causing performance degradation. In this paper, we propose a novel **A**daptive **M**ulti-dimensional **S**tructured **C**ompression (AMSC) method that simultaneously learns the minimal depth, the minimal width, and network parameters under the strategy that prioritizes depth compression. Specifically, based on the regularization technique, AMSC incorporates layer- and filter- specific information into the penalty in order to adaptively identify and eliminate redundant depth and width in terms of the importance and size of each layer and filter. It integrates compression and training processes together without pruning. Consequently, the proposed method enables adaptive structure reduction from the initial configuration to a structure necessary that minimizes the generalization error. Rigorous theoretical evidence is provided in terms of the consistency of AMSC in achieving minimal network depth and width. To the best of our knowledge, this is the first study that offers a theoretical guarantees in structure selection. Extensive experiments on CIFAR-10/100 and ImageNet datasets demonstrate our method not only achieves state-of-the-art compression performance in terms of FLOPs and total parameters, but also preserves competitive classification accuracy. For example, AMSC enhances the accuracy of ResNet56 on CIFAR-10 from 93.37% to 93.71%, while simultaneously reducing calculations by 58.63% and parameters by 44.71%.

## 1    INTRODUCTION

Deep neural networks (DNNs) have shown significant advancements across various domains (Krizhevsky et al., 2012; Kenton & Toutanova, 2019). However, their extensive parameterization presents several challenges. First, training DNNs with numerous parameters necessitates a massive amount of samples, which is often impractical, particularly in specialized domains. Second, the large number of parameters in DNNs can theoretically increase statistical error, thereby potentially reducing the networks' overall generalization ability (Jiao et al., 2023; Tan et al., 2024). Third, DNNs with substantial parameter counts demand significant storage space and exhibit slower runtime (Chen & Zhao, 2019; Wu et al., 2023), hindering their deployment on resource-constrained edge devices, such as mobile phones, robotics, drones and smart watches.

To address these issues, numerous model compression approaches have been explored (Frankle & Carbin, 2018; Tan et al., 2020; Gou et al., 2021). A key technique among them is network pruning which involves removing redundant parameters and connections, and is categorized into either unstructured (Han et al., 2015; Frankle & Carbin, 2018; Sun et al., 2023) or structured (Li et al., 2017; Yu & Xiang, 2023; Chen & Zhao, 2019; Yu et al., 2022). Unstructured pruning deletes individual weights (weight-level) yet often struggles to achieve substantial speedup without specialized libraries or hardware (Han et al., 2015; Lin et al., 2019b). In

contrast, structured pruning eliminates redundant filters or layers to reduce network width or depth, and is better suited for acceleration on regular devices (Lin et al., 2020).

Current structured pruning methods mainly target a single dimension, such as pruning filters in CNNs or self-attention heads in Transformers to reduce width, or removing layers and blocks to decrease depth. However, focusing on only one dimension can lead to sub-optimal compression when multiple dimensions exhibit redundancy, particularly when prior knowledge about which dimensions are truly redundant is limited, and forcibly pruning a non-redundant dimension not only fails to achieve significant compression but also greatly reduces accuracy (Wang et al., 2021b). Therefore, data-driven multi-dimensional pruning is crucial for effective and precise compression, allowing faster model acceleration without sacrificing quality.

Current multi-dimensional structured pruning methods mainly focus on the allocation of pruning weights in each dimension (Wang et al., 2021b) and algorithms for identifying redundant structures (Wen et al., 2016; Lin et al., 2019b). While these approaches offer powerful tools for model compression, the literature lacks systematic research on the following issues:

(1) Given a network architecture and a specific compression strategy, is there a sub-network with minimal depth and minimal width that maintains the minimum generalization error?

(2) If such a sub-network exists, how to design an algorithm to accurately identify and train it?

In this paper, we aim to address the aforementioned questions. Specifically, since wide networks are easier to optimize (Glorot & Bengio, 2010; Srivastava et al., 2015) and can typically be executed in parallel that facilitating acceleration (Kim et al., 2023; 2024), we adopt a strategy that prioritizes depth compression. We propose a novel penalty-based **A**daptive **M**ulti-dimension **S**tructured **C**ompression (AMSC) method to adaptively and simultaneously compress both the depth and width while estimating network parameters. The crucial point of AMSC in depth compression lies in observing that layers performing identity mapping can be removed without impacting the network's architecture or performance. Then, by imposing an identity penalty on layer parameters, AMSC adaptively identifies redundant layers. Furthermore, it narrows network width by penalizing width units such as filters in CNNs or heads in Transformers.

Unlike previous methods, AMSC seamlessly integrates the compression and training processes, enabling simultaneous learning of network structure and parameters. Consequently, AMSC offers adaptive structure reduction from the initial structure to a structure necessary to minimize generalization error. The main contributions of this paper are as follows:

(i) We first investigate the relationship between accuracy and network structure across diverse architectures. Figure 2 (a-b) illustrates that accuracy (in blue) initially increases and then decreases with reduced network depth (in green), yet it remains above the baseline (in black), even for networks with substantially fewer layers. A similar trend is observed in the process of decreasing the network width, as shown in Figure 2 (c-d). These results reveal that tighter sub-networks within the original architecture can achieve greater accuracy than can the full model. Moreover, during the structure-compressing phase, the accuracy reaches a plateau and then sharply declines if the reduction continues. This phenomenon emphasizes the existence of a minimal depth and width to ensure generalization error.

(ii) To identify the minimal structure, we propose the AMSC method for compressing various DNNs using a penalty technique. Unlike existing penalty-based multi-dimensional compression methods (Wen et al., 2016; Lin et al., 2019b), which equally penalize each layer or filter and often compress those with fewer parameter counts, our penalty integrates both the importance and parameter counts of layers and filters into the weights. This allows AMSC to achieve precise and effective compression in terms of FLOPs and parameter counts while maintaining higher accuracy, as shown in Table 1.

(iii) We provide rigorous theoretical evidence that the proposed AMSC can achieve the minimal depth and width with performance guarantees, as shown in Theorem 4.1. To our knowledge, this is the first study

offering theoretical guarantees for structure selection. Additionally, we explain how equally penalizing each layer and filter can lead to improper compression, potentially violating Assumption 4.1, which is necessary for selection consistency, as illustrated in Figure 1.

We conduct extensive experiments on the CIFAR-10/100 and ImageNet datasets, demonstrating that our proposed AMSC achieves state-of-the-art compression performance, while maintaining competitive classification accuracy compared to existing methods, as detailed in Tables 1 and 5. For instance, AMSC increases the accuracy of ResNet56 on CIFAR-10 from 93.37% to 93.71%, while simultaneously reducing computations by 58.63% and parameters by 44.71%. Moreover, the structure identified by AMSC remains consistent across different initial network architectures, further validating our theoretical findings.

## 2 RELATED WORKS

**Unstructured and single-dimensional structured pruning.** Neural network pruning initially occurs at the weight-level (Han et al., 2015; Frankle & Carbin, 2018; Chen et al., 2020b; Sun et al., 2023), achieving sparsity by eliminating unimportant weights. However, the resultant unstructured sparsity achieves acceleration only under specific libraries (e.g., cuSPARSE), which is restricted on mobile devices (Wang et al., 2021b). Recently, single-dimensional structured pruning emerges, which focuses on reducing either the width (He et al., 2017; Ding et al., 2021a; Michel et al., 2019) or depth (Chen & Zhao, 2019; Jordao et al., 2020; Kim et al., 2023; 2024) of DNNs. These methods primarily follow two research trajectories. The first line (Hu et al., 2016; He et al., 2019; Lee et al., 2019; Wang et al., 2019; Yu et al., 2022) assesses the importance of network modules, pruning those deemed least important. The second line (Wen et al., 2016; Zhu et al., 2018; Lin et al., 2019a; Wang et al., 2020; Wu et al., 2023) introduces a sparsity penalty into the objective function to learn compact models during the training phase. Although they all receive great success in pruning DNNs, their focus on a single dimension limits compression potential, especially when multiple dimensions exhibit redundancy, which is often the case.

**Multi-dimensional pruning.** To enhance compression efficacy, several methods explore the multi-dimension pruning and can generally be divided into two groups. The first group (Wen et al., 2016; Lin et al., 2019b) reduces both the width and depth of models by imposing extra penalty terms on the network structures. However, these methods equally penalize all network components, such as layers and filters, ignoring the importance and parameter counts of different components. This can lead to inadequate compression, as demonstrated by GAL and SSL in Table 1. The second group (Wang et al., 2021b; Yu et al., 2022) assigns pruning budgets to different dimensions but prunes network based on traditional single-dimensional methods. Particularly, Wang et al. (2021b) formulate the allocation issue as an optimization problem, and establish the optimization targets by searching networks with varying depths and widths, making the allocation process time-consuming. Yu et al. (2022) adopts a sequential pruning strategy, first pruning width, then depth. However, both methods separate pruning from training, disrupting established correlations within the network and resulting in performance degradation (Ding et al., 2021a;b). Additionally, none of the aforementioned methods are theoretically supported. In contrast, the proposed AMSC incorporates components-specific information into the penalty to adaptively identify and eliminate redundant components, and integrates compression and training processes together without pruning, yielding better compression performance. Furthermore, we provide rigorous theoretical evidence that under mild conditions, AMSC can automatically identify minimal network depth and width given depth-first compression strategy.

**Neural Architecture Search (NAS).** The objective of NAS (Pham et al., 2018; Tan & Le, 2019; Gao et al., 2020b; Guo et al., 2021; 2023b) is to identify a well-designed architecture by searching the options and connections given a computational budget. This goal is similar with pruging. However, their operational frameworks are very different: NAS constructs models from scratch, while pruning reduce the scale of an existing model. Thus, despite several NAS algorithms (Tan & Le, 2019; Han et al., 2020) strive to optimize multiple dimensions (e.g., depth, width) of a model, they are not directly applicable to the pruning domain.

# 3 ADAPTIVE MULTI-DIMENSION COMPRESSION

A neural network with $L-1$ middle layers is a collection of mappings $f$ of the form $f(x, \theta) = f_L \circ f_{L-1} \circ f_{L-2}... \circ f_0(x)$, where $\theta = \{\theta_l\}_{l=0}^L$ are the parameters, and $f_1 \circ f_0(x) = f_1(f_0(x))$ represents the composition of two functions $f_1$ and $f_0$. Then the underlying mapping in the $l$-th layer can be formulated as:

$$z_l = f_l(z_{l-1}, \theta_l),\ 1 \le l \le L-1, \tag{1}$$

where $z_l$ is the output of the $l$-th layer, and $f_l(\cdot, \theta_l)$ is the neural network with parameters $\theta_l$. In practical DNNs implementations, the depth is usually given in advance, and changing the depth requires retraining a new network. This limitation hinders the dynamic adjustment between network training and depth. To address this, we propose a new strategy based on a key observation, in which if $z_l = z_{l-1}$, the $l$-th layer is redundant. More specifically, the layer performing identity mapping can be removed without disrupting the well-established correlations within the network or compromising network performance. To determine whether $z_l = z_{l-1}$, we rewrite model (1) as

$$z_l = h_l(z_{l-1}, \theta_l) + z_{l-1},\ 1 \le l \le L-1. \tag{2}$$

Then identifying whether $z_l = z_{l-1}$ is transformed into $h_l(z_{l-1}, \theta_l) = 0$. Based on the network expression for $h_l$, the identification of redundant layers can be further reformulated as the problem of determining whether the parameters $\theta_l$ are zero, which can be implemented by imposing a group penalty on $\theta_l$.

Network width typically refers to the number of filters in CNNs or the number of attention heads in Transformers. For simplicity, we uniformly refer to these as "filters" when discussing width units. To decrease the width, we further impose a group penalty on $\theta_{l,j}$, where $\theta_{l,j}$ is the $j$-th filter of the $l$-th layer. As a result, our objective in a dataset $\{X_i, y_i\}_{i=1}^n$ with $n$ samples ends up with:

$$\hat{\theta} = \arg\min_{\theta} \mathcal{L}(\theta) := \frac{1}{n} \sum_{i=1}^n \mathcal{L}_{target}(y_i, f(X_i, \theta)) + \lambda_0 Q_d(\theta) + \lambda_1 Q_w(\theta) \tag{3}$$

where $\mathcal{L}_{target}(\cdot, \cdot)$ is the loss function for specific targets, such as mean square error (MSE) in regression and cross-entropy (CE) loss in classification; the penalty $Q_d(\theta) = \sum_{l=1}^{L-1} \lambda(l)\|\theta_l\|_2$ and $Q_w(\theta) = \sum_{l=1}^{L-1} \sum_{j=1}^{n_l} \lambda(l,j)\|\theta_{l,j}\|_2$ are used to identify redundant depth and width; $\lambda_0$ and $\lambda_1$ are the penalty intensity to balance the target loss and the network architecture. As $\lambda_0$ and $\lambda_1$ increase, the number of zeros in $\{\theta_l | 1 \le l \le L-1\}$ and $\{\theta_{l,j} | 1 \le l \le L-1, 1 \le j \le n_l\}$ increases, resulting in a shallow and narrow neural network architecture. Hence the automatic selection of depth and width can be accomplished by tuning the parameters $\lambda_0$ and $\lambda_1$.

$\lambda(l)$ and $\lambda(l,j)$, the weight of the $l$-th layer and the $j$-th filter in $l$-th layer, play crucial roles in identifying redundant layers and filters. Setting $\lambda(l) = 1$ and $\lambda(l,j) = 1$, penalizing each layer and filter equally, might overlook the sequence of layers and the variation in parameter count among layers and filters, potentially leading to compressing layers and filters with fewer parameters. Taking depth as an example, in networks such as ResNet, earlier layers typically have fewer parameters, serve as fundamental components for later layers and should be retained. Therefore, the approach with $\lambda(l) = 1$ may lead to inadequate compression and suboptimal performance, as demonstrated by SSL in Table 2 and Figure 3. Here, we hence design the weight based on two considerations: the importance of each layer and filter, and the parameter counts present in them. In particular, we set the weights as

$$\lambda(l) = \sqrt{q_l}/\|\hat{\theta}_l\|_2,\ \lambda(l,j) = \sqrt{q_{l,j}}/\|\hat{\theta}_{l,j}\|_2, \tag{4}$$

where $q_l$ and $q_{l,j}$ are the number of parameters in the $l$-th layer and the $j$-th filter of the $l$-th layer, respectively. $\hat{\theta}_l$ and $\hat{\theta}_{l,j}$ are estimators for $\theta_l$ and $\theta_{l,j}$, which can be obtained from the pre-trained model. This choice of $\lambda(l)$ and $\lambda(l,j)$ has several intriguing features. First, under the commonly used assumption that $\|\theta\|_\infty < B$ for

some $0 < B < +\infty$ in neural network literature (Chen et al., 2020a), $\|\theta_l\|_2 \leq \sqrt{q_l}B$ and $\|\theta_{l,j}\|_2 \leq \sqrt{q_{l,j}}B$. Then, the weights $\sqrt{q_l}$ and $\sqrt{q_{l,j}}$ serve as adaptive tuning parameters for shrinking $\theta_l$ and $\theta_{l,j}$ toward zero based on their respective parameter counts. This is important because parameter counts can vary significantly across layers and filters. Second, DNNs are highly unidentifiable (Fukumizu, 2003) even with specified width, depth, and loss. Given width, depth, and loss, the AMSC method, using the weight $\sqrt{q_l}$ and $\sqrt{q_{l,j}}$, tends to compress layers and filters with more parameters, resulting in a simpler network with guaranteed performance. Third, the magnitude of $\|\hat{\theta}_l\|_2$ and $\|\hat{\theta}_{l,j}\|_2$ reflect the importance of the $l$-th layer and the $j$-th filter of the $l$-th layer, respectively. $1/\|\hat{\theta}_l\|_2$ and $1/\|\hat{\theta}_{l,j}\|_2$ assign more weights to layers and filters with lower norm values, guiding AMSC to aggressively compress the less important components.

## 4 CONSISTENCY IN ARCHITECTURE SELECTION

In this section, we theoretically demonstrate that the proposed AMSC can identify the minimal depth and width in terms of selection consistency, given depth-first compression strategy. The rationale for prioritizing depth-wise compression is that reducing depth facilitates model optimization (Glorot & Bengio, 2010; Srivastava et al., 2015) and acceleration (Kim et al., 2023; 2024) compared to reducing width. Denote the optimal parameter set that minimize the generalization error as $\Theta^* = \{\theta^* : \theta^* \in \arg\min_\theta \mathbb{E}_{(X,y)\sim\mu}\mathcal{L}_{target}(y, f(X, \theta))\}$, where $\mu$ is the population distribution of samples. Since DNNs are highly unidentifiable, we first consider depth and width selection consistency given another component. Particularly, given width $W = w$, the minimal depth that maintains the minimum generalization error is defined as $l_\theta(w) = \min_{l^*}\{l^* : l^* = \mathrm{dep}(\theta^*), \theta^* \in \Theta^*, W = w\}$, where $\mathrm{dep}(\theta^*) = \sum_{l=1}^{L-1} \mathbf{1}_{\{\theta_l^* \neq 0\}}$ is the depth of $\theta^*$. Hereafter, we drop $w$ for brevity if the $w$ is the initial width. Similarly, given $L = l$, the minimal width that maintains the minimum generalization error is defined as $w_\theta(l) = \min_{w^*}\{w^* : w^* = \mathrm{wid}(\theta^*), \theta^* \in \Theta^*, L = l\}$, where $\mathrm{width}(\theta^*) = \sum_{l=1}^{L-1}\sum_{j=1}^{n_l} \mathbf{1}_{\{\theta_{l,j}^* \neq 0\}}$ is the width of $\theta^*$. Then, we define that the estimation of $f$ achieves consistency in structure selection if $\mathbb{P}(\mathrm{dep}(\hat{\theta}) = l_\theta) \to 1$, $\mathbb{P}(\mathrm{wid}(\hat{\theta}) = w_\theta(l_\theta)) \to 1$ as $n \to \infty$. In the above consistency definition, we identify the minimal depth under initial width $w$ and access the minimal width given the minimal depth. It should be noted that further depth compression on $\mathrm{dep}(\hat{\theta})$ is impossible because $\mathrm{wid}(\hat{\theta})$ is less than the initial width $w$. Thus, $\mathrm{dep}(\hat{\theta})$ then $\mathrm{wid}(\hat{\theta})$ represent the minimum depth and width under the current compression strategy. The following assumptions are required to establish their consistency.

**Assumption 4.1.** *For any $\theta_1^*, \theta_2^* \in \Theta^*$, $Q_d(\theta_1^*) \leq Q_d(\theta_2^*)$ implies $dep(\theta_1^*) \leq dep(\theta_2^*)$; and $Q_w(\theta_1^*) \leq Q_w(\theta_2^*)$ implies $wid(\theta_1^*) \leq wid(\theta_2^*)$.*

Assumption 4.1 requires the penalty terms $Q_d(\theta)$ and $Q_w(\theta)$ to have the ability to identify redundant layers and filters. In section 5.3.1, we empirically shows the monotonic relationship between $\mathrm{dep}(\theta)$ and $Q_d(\theta)$, as well as between $\mathrm{wid}(\theta)$ and $Q_w(\theta)$. Meanwhile, we also demonstrate that some commonly used penalties (Wen et al., 2016) may violate Assumption 4.1, resulting in inappropriate compression.

**Assumption 4.2.** *The loss function $\mathcal{L}_{target}(\theta)$ is a sub-analytic function of $\theta$.*

The loss functions such as CE loss, and DNNs with activation functions ReLU and GeLU (Hendrycks & Gimpel, 2016), are sub-analytic functions (Bolte et al., 2006), Assumption 4.2 is typically satisfied in practice.

**Assumption 4.3.** *For any two $\theta_1^*, \theta_2^* \in \Theta^*$, $\|\theta_1^* - \theta_2^*\|_2 \leq M_b < +\infty$.*

Assumption 4.3 focuses on the bounded difference between any two parameters in $\Theta^*$. It is a relaxation of the assumption that the $L_1$ norm of any parameter is bounded, a condition frequently required in the neural network literature (Chen et al., 2020a).

**Theorem 4.1.** *Suppose that Assumptions 4.1, 4.2 and 4.3 hold. Let $\hat{\theta}$ be the estimator of equation 3, if $\lambda_0 = o_p(1)$, $\lambda_1 = o_p(\lambda_0)$ and the statistical error $S_n = o_p(\lambda_1)$, we deduce that*

$$\mathbb{P}(dep(\hat{\theta}) = l_\theta) \to 1, \ \mathbb{P}(wid(\hat{\theta}) = w_\theta(l_\theta)) \to 1, \ d(\hat{\theta}, \Theta^*)\hat{=}\min_{\theta^*\in\Theta^*}\|\hat{\theta} - \theta^*\|_2 = o_p(1). \tag{5}$$

Theorem 4.1 demonstrates that AMSC can identify the minimal structure with appropriate choices of $\lambda_0$ and $\lambda_1$. The condition $\lambda_1 = o_p(\lambda_0)$ is required to ensure that the selection of depth is independent of the selection of width. The requirement that $\lambda_0$ and $\lambda_1$ exceed the statistical error $S_n$ is to prevent cases where redundant structures remain uncompressed due to randomness.

Mathematically, denote the class of neural network is $\mathcal{F}$. The statistical error $S_n$ can be bounded by the pseudo dimension of $\mathcal{F}$ (Jiao et al., 2023), denoted by $\text{Pdim}(\mathcal{F})$. In particular, if both the architecture and activation functions within $\mathcal{F}$ remain fixed, it follows that $\text{Pdim}(\mathcal{F}) = \text{VCdim}(\mathcal{F})$ (Bartlett, 1996), where $\text{VCdim}(\mathcal{F})$ is the Vapnik-Chervonenkis (VC) dimension of $\mathcal{F}$, which can be further bounded by width, depth and the number of parameters of $\mathcal{F}$ (Jiao et al., 2023; Bartlett et al., 2019). There are several existing results on the statistical error. Chen et al. (2020a) demonstrate that it scales as $\mathcal{O}_p(n^{-\frac{\beta}{2\beta+d}})$ where $\beta$ is the smoothness index of true function class and $d$ is the input dimension. This rate can be further refined to $\mathcal{O}_p(n^{-\frac{\beta}{2\beta+d^*}})$, where $d^*$ is the intrinsic dimension of data (Nakada & Imaizumi, 2020).

The detailed proof of Theorem 4.1 can be found in Appendix A. The proof includes two parts: convergence to the optimal parameter set and structure consistency, where the first is achieved by the Lojasiewicz inequality (Ji et al., 1992; Colding & Minicozzi, 2014; Bolte et al., 2006), Young's inequality and Assumption 4.2, and the second part can be obtained by the property of $\hat{\theta}$ and Assumption 4.1 and 4.3.

## 5 EXPERIMENTS

### 5.1 IMPLEMENT DETAILS AND EXPERIMENTAL SETTINGS

**Training procedures.** We implement the proposed AMSC by optimizing equation 3, with the details provided in Algorithm 1 of Appendix B.1.

**Datasets.** We use three popular datasets to test the proposed AMSC: CIFAR-10/100 (Krizhevsky et al., 2009) and ImageNet ILSVRC 2012 (Deng et al., 2009). These datasets differs in image resolution (from $32 \times 32$ to $224 \times 224$), number of classes (10 to 1000), and dataset size (50K to 1M). For all datasets, we apply common augmentation techniques such as symmetric padding, random cropping, and horizontal flipping, in line with standard practices (He et al., 2016; Huang et al., 2017).

**Architectures.** Our experiment spans various architecture, including VGGs (Simonyan & Zisserman, 2015), ResNets (He et al., 2016), DenseNets (Huang et al., 2017) and DeiTs (Touvron et al., 2021). For depth compression, we follow standard settings (Wang et al., 2019; Zhang et al., 2024) that designate layers in VGGs, blocks in ResNets and DenseNets, attention and FFN layers in DeiT as compression units. As VGGs lack skip connections, we modify the original architecture to maintain connectivity when implementing AMSC and the details are available in Appendix B.2. For width compression, we take convolutional filters in CNNs, and heads for attention layers and neurons for FFN layers in Transformers as compression units.

**Training settings.** The training settings for all architectures on different datasets follow commonly used protocols (He et al., 2016; Huang et al., 2017; Touvron et al., 2021). Detailed training configurations, along with the selection strategies for $\lambda_0$ and $\lambda_1$ in AMSC, are provided in Appendix B.3.

**Evaluation protocols.** We evaluate the compression ratios by floating-point operations (FLOPs) and parameter counts (Params.). To minimize bias due to differing experimental conditions, we adopt the approach from (He et al., 2018; Gao et al., 2020a; Wu et al., 2023) by using the relative accuracy increase to the benchmark model to investigate model performance. The compression ratios, accuracy, and corresponding baselines for other methods are directly taken from the original studies.

### 5.2 RESULTS AND ANALYSIS

**Results on CIFAR**. We analyze the performance of the proposed AMSC on CIFAR-10, comparing it to several popular CNNs, including ResNet-56, ResNet-110, and DenseNet-40. The results are shown in Table 1

Table 1: Performance comparisons for various architectures on CIFAR-10 and ImageNet. Pruned and Acc.↑ denote pruned accuracy and relative accuracy increase, respectively. W and D indicate whether the model will be pruned along depth and width, respectively. The best and second best scores are highlighted as **bold** and underlined, respectively.

| Dataset | Architecture | Methods | W | D | Baseline(%) | Pruned(%) | Acc↑(%) | FLOPs(M/B) | Params.(M) |
|---------|-------------|---------|---|---|-------------|-----------|---------|------------|------------|
| CIFAR-10 | ResNet56 | GAL(Lin et al., 2019b) | ✓ | ✓ | 93.26 | 93.38 | 0.12 | 78.74 | 0.75 |
| | | DLP(Jordao et al., 2020) | | ✓ | - | - | -0.82 | 65.80 | 0.52 |
| | | TDPF(Wang et al., 2021b) | ✓ | ✓ | 93.69 | 93.76 | 0.09 | 63.50 | 0.51 |
| | | HRank(Lin et al., 2020) | ✓ | | 93.26 | 93.17 | -0.09 | 62.72 | 0.49 |
| | | SANP(Gao et al., 2023) | ✓ | | 93.49 | 93.81 | 0.32 | 60.24 | - |
| | | LPSR(Zhang & Liu, 2022) | | ✓ | 93.21 | 93.40 | 0.19 | 60.10 | 0.47 |
| | | SSL(Wen et al., 2016) | | ✓ | 93.37 | 93.25 | -0.12 | 59.79 | 0.50 |
| | | ELC(Wu et al., 2023) | | ✓ | 93.45 | 93.66 | 0.21 | 58.30 | - |
| | | AMSC(Ours) | ✓ | ✓ | 93.37 | 93.71 | **0.34** | **51.91** | 0.47 |
| | ResNet110 | DBP(Wang et al., 2019) | | ✓ | 93.97 | 93.61 | -0.36 | 141.90 | - |
| | | GAL(Lin et al., 2019b) | ✓ | ✓ | 93.50 | 92.55 | -0.95 | 130.20 | 0.95 |
| | | DLP(Jordao et al., 2020) | | ✓ | - | - | -0.25 | 129.70 | 1.02 |
| | | ELC(Wu et al., 2023) | | ✓ | 93.60 | 94.07 | **0.47** | 92.30 | - |
| | | HRank(Lin et al., 2020) | ✓ | | 93.50 | 92.65 | -0.85 | 79.30 | 0.70 |
| | | DECORE(Alwani et al., 2022) | ✓ | | 93.50 | 92.71 | -0.79 | 58.16 | 0.35 |
| | | AMSC(Ours) | ✓ | ✓ | 93.51 | 92.73 | -0.78 | **54.71** | **0.34** |
| | DenseNet40 | DBP(Wang et al., 2019) | | ✓ | 94.59 | 94.02 | -0.57 | 159.25 | 0.43 |
| | | DECORE(Alwani et al., 2022) | ✓ | | 94.81 | 94.04 | -0.77 | 128.13 | 0.37 |
| | | GAL(Lin et al., 2019b) | ✓ | ✓ | 94.81 | 93.53 | -1.28 | 128.11 | 0.45 |
| | | DHP(Li et al., 2020) | ✓ | | 94.74 | 93.94 | -0.80 | 112.06 | 0.68 |
| | | HRank(Lin et al., 2020) | ✓ | | 94.81 | 93.68 | -1.13 | 110.15 | 0.48 |
| | | AMSC(Ours) | ✓ | ✓ | 94.07 | 93.93 | **-0.14** | **103.43** | 0.33 |
| ImageNet | ResNet34 | Taylor (Molchanov et al., 2019) | ✓ | | 73.31 | 72.83 | -0.48 | 2.83 | 17.20 |
| | | FPGM (He et al., 2019) | ✓ | | 73.92 | 72.63 | -1.29 | **2.16** | - |
| | | LPSR(Zhang & Liu, 2022) | | ✓ | 73.31 | 72.63 | -0.68 | 2.52 | **14.35** |
| | | ELC (Wu et al., 2023) | | ✓ | 74.02 | 73.79 | **-0.23** | 2.43 | - |
| | | AMSC(Ours) | ✓ | ✓ | 73.31 | 72.93 | -0.38 | 2.27 | 16.61 |
| | ResNet50 | GAL(Lin et al., 2019b) | ✓ | ✓ | 76.15 | 71.95 | -4.20 | 2.33 | 21.20 |
| | | HRank(Lin et al., 2020) | ✓ | | 76.15 | 74.98 | -1.17 | 2.30 | 16.15 |
| | | AKECP(Zhang et al., 2021) | ✓ | | 76.52 | 76.20 | -0.32 | 2.29 | **15.16** |
| | | Greg-2(Wang et al., 2021a) | ✓ | | 76.13 | 75.36 | -0.77 | **1.77** | - |
| | | GFP (Liu et al., 2021) | ✓ | | 76.79 | 76.42 | -0.37 | 2.04 | - |
| | | DepGraph (Fang et al., 2023) | ✓ | ✓ | 76.15 | 75.83 | **-0.32** | 1.99 | - |
| | | AMSC(Ours) | ✓ | ✓ | 76.15 | 75.53 | -0.62 | 1.85 | 16.84 |
| | DeiT-tiny | OPTIN-$\beta$ (Khaki & Plataniotis, 2024) | ✓ | | 72.20 | 67.51 | -4.69 | 1.10 | - |
| | | SSP (Michel et al., 2019) | ✓ | | 72.20 | 68.60 | -3.60 | 1.00 | 4.20 |
| | | S2ViTE (Chen et al., 2021) | ✓ | | 72.20 | 70.10 | -2.10 | 1.00 | 4.20 |
| | | SPViT (He et al., 2024) | ✓ | | 72.20 | 70.70 | -1.50 | 1.00 | 4.80 |
| | | P6 (Liu et al., 2024) | | ✓ | 72.20 | 70.30 | -1.90 | 0.90 | **3.80** |
| | | AMSC(Ours) | ✓ | ✓ | 72.20 | 70.70 | **-1.50** | **0.87** | 4.60 |

(Top block). As observed, maintaining similar accuracy levels, width-based compression methods offer lower compression rates than depth-based methods for ResNet56. However, this trend reverses for ResNet-110. This phenomenon highlights that the significance of depth and width varies across different networks; in other words, a single-dimensional compression strategy-either width or depth- does not universally apply to all network architectures. In contrast, the proposed AMSC achieves a higher relative accuracy improvement, requires fewer FLOPs, and utilizes fewer total parameters compared to the single-dimensional compression methods, emphasizing the advantages of multi-dimensional compression. Moreover, compared to existing multi-dimensional compression methods such as GAL (Lin et al., 2019a) and TDPF (Wang et al., 2021b), which either treat all components equally or rely on traditional single-dimension pruning techniques, AMSC achieves significantly enhanced compression efficiency by incorporating structure-specific information into its penalty and integrating compression with training. Additionally, the architectures derived from both ResNet56 and ResNet110 via AMSC achieve similar accuracy, FLOPs and the number of parameters. This implies the existence of a minimal structure that remains consistent across different initial architectures and confirms the consistency in the structure selection of AMSC, as established in Theorem 4.1. We also conduct experiments for VGG16 on CIFAR10, and ResNet56, VGG16 and VGG19 on CIFAR-100. The experimental results are presented in Appendix B.4, yielding similar conclusions to those obtained for CIFAR-10.

**Results on ImageNet**. We also conduct experiments for ResNets and DeiTs on the challenging ImageNet dataset. As demonstrated in Table 1 (Bottom block), AMSC achieves the fewer FLOPs with a competitive accuracy for both ResNet34 and DeiT-tiny. However, for ResNet34, LPSR maintains a fewer parameters than AMSC. To see the inconsistency, we display the compressed ResNet34 in Figure 3 (Bottom). LPSR prunes 5 blocks, including a critical one in the penultimate position which has the highest parameter count in the network and is crucial for the network performance (Zeiler & Fergus, 2014). Therefore, pruning this block leads to LPSR having significantly reduced parameter counts but at the cost of a substantial performance decrease (-0.68%) and a slight reduction in FLOPs (2.52B). Conversely, AMSC prunes 6 blocks primarily within the middle layers, which typically exhibit weaker feature extraction capabilities (Nguyen et al., 2020) and possess fewer parameters, resulting in a smaller accuracy decrease (-0.32%) and fewer FLOPs (2.27B). For ResNet50, AMSC exhibits a behavior similar to that observed in ResNet34, focusing on compressing the middle layers, which are relatively less important. Hence, AMSC results in highly competitive compression performance. For Deit-Tiny, AMSC compresses two attention layers that are high in FLOPs yet low in parameters, resulting in a model with fewer FLOPs (0.87B), higher accuracy (70.70%), and increased parameter counts (4.60M). These result implies that AMSC targets less critical structures, achieving sufficient compression with performance guarantees.

## 5.3 ABLATION STUDY AND DISCUSSION

### 5.3.1 AN EMPIRICAL VERIFICATION OF ASSUMPTION 4.1

Assumption 4.1 requires the penalty term $Q_d(\theta)$ and $Q_w(\theta)$ to have the abilities to identify redundant structures. For clarity, we compare the proposed penalties $Q_d(\theta) = \sum_{l=1}^{L-1} \frac{1}{\|\hat{\theta}_l\|_2} \sqrt{q_l} \|\theta_l\|_2$ and $Q_w(\theta) = \sum_{l=1}^{L-1} \sum_{j=1}^{n_l} \frac{1}{\|\hat{\theta}_{l,j}\|_2} \sqrt{q_{l,j}} \|\theta_{l,j}\|_2$ with the penalties $\tilde{Q}_d(\theta) = \sum_{l=1}^{L-1} \|\theta_l\|_2$ and $\tilde{Q}_w(\theta) = \sum_{l=1}^{L-1} \sum_{j=1}^{n_l} \|\theta_{l,j}\|_2$ in SSL (Wen et al., 2016). Removing a network component with a high parameter count can significantly decreases $\tilde{Q}_d(\theta)$ and $\tilde{Q}_w(\theta)$, but the reduction in depth or width can be minor. Therefore, $\tilde{Q}_d(\theta)$ and $\tilde{Q}_w(\theta)$ may not fulfill Assumption 4.1. In contrast, $Q_d(\theta)$ and $Q_w(\theta)$ assigns weights to each layers and filters based on their importance and parameter counts, aggressively compressing less important layers and filters. This makes $Q_d(\theta)$ and $Q_w(\theta)$ more likely to satisfy Assumption 4.1. The empirical verification of the above analysis can be found in Figure 1. Based on the results from ResNet56 and DenseNet40 on CIFAR-10, and DeiT-tiny on ImageNet, Figure 1 (a-b) illustrates a monotonic relationship between $\text{dep}(\theta^*)$ and $Q_d(\theta^*)$ (in blue). However, this monotonicity between $\text{dep}(\theta^*)$ and $\tilde{Q}_d(\theta^*)$ (in orange) is not so significant. Similar results are also observed between $\text{width}(\theta^*)$ and $Q_w(\theta^*)$ ( $\tilde{Q}_w(\theta^*)$) in Figure 1 (c-d). More verification of Assumption 4.1 can be found in Appendix B.5.

### 5.3.2 THE INFLUENCE OF $\lambda_0$ AND $\lambda_1$

We demonstrate the influence of the penalty parameters $\lambda_0$ and $\lambda_1$ in Figure 2 based on the results from ResNet56 and DenseNet40 on CIFAR-10, and DeiT-tiny on ImageNet. In Figure 2 (a-b), as $\lambda_0$ increases, network depth (in green) decreases, and accuracy (in blue) initially increases and then decreases, but it stays above the baseline even for networks with significantly small depth. This result reveals that shallower sub-networks within the original architecture can achieve greater accuracy compared to the full model. This occurs because AMSC selects the smallest network that ensures generalization error, resulting in substantially decreased statistical error while maintaining approximation error. Moreover, during the decreasing phase of depth, accuracy reaches a plateau and then sharply declines if the depth is continuously reduced. This phenomenon emphasizes the existence of minimal depth which ensures the generalization error and AMSC's capability to identify and respond to it. Similar patterns are also observed in width reduction as $\lambda_1$ increases, as shown in Figure 2 (c-d). More results about the influence of $\lambda_0$ and $\lambda_1$ can be found in Appendix B.6.

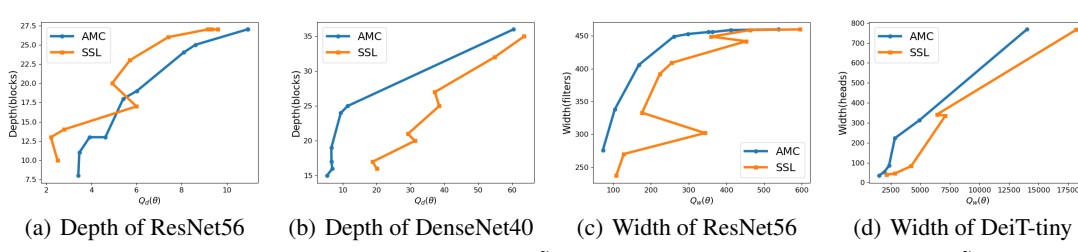

(a) Depth of ResNet56    (b) Depth of DenseNet40    (c) Width of ResNet56    (d) Width of DeiT-tiny

Figure 1: Relationship between $\text{dep}(\theta^*)$ and $Q_d(\theta^*)$ ($\tilde{Q}_d(\theta^*)$) as well as $\text{wid}(\theta^*)$ and $Q_w(\theta^*)$ ($\tilde{Q}_w(\theta^*)$) for any $\theta^* \in \Theta^*$, where $Q(\theta) = \sum_{l=1}^{L-1} \frac{1}{\|\hat{\theta}_l\|_2} \sqrt{q_l} \|\theta_l\|_2$ and $Q_w(\theta) = \sum_{l=1}^{L-1} \sum_{j=1}^{n_l} \frac{1}{\|\hat{\theta}_{l,j}\|_2} \sqrt{q_{l,j}} \|\theta_{l,j}\|_2$ are the proposed penalties as well as $\tilde{Q}_d(\theta) = \sum_{l=1}^{L-1} \|\theta_l\|_2$ and $\tilde{Q}_w(\theta) = \sum_{l=1}^{L-1} \sum_{j=1}^{n_l} \|\theta_{l,j}\|_2$ are the penalties used in SSL (Wen et al., 2016). The accuracy of each point meets or exceeds the baseline and varies by less than $1\%$ across different points within the same architecture. Therefore, these points can be approximately considered as elements of $\Theta^*$.

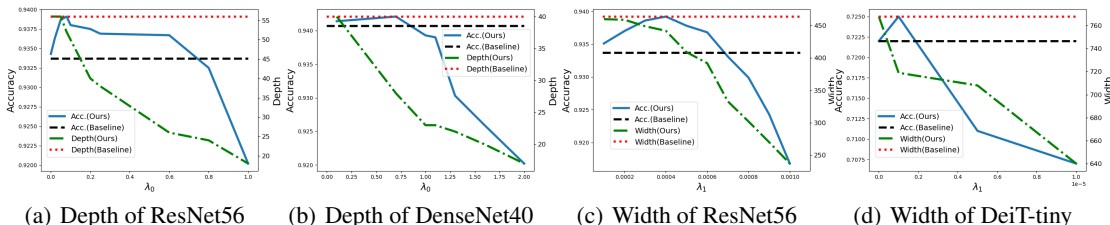

(a) Depth of ResNet56    (b) Depth of DenseNet40    (c) Width of ResNet56    (d) Width of DeiT-tiny

Figure 2: The accuracy and depth/width of different architectures vary with the penalty parameters $\lambda_0$ and $\lambda_1$, In each figure, the left Y-axis denotes the accuracy and the right Y-axis denotes the depth (width).

### 5.3.3 CHOICE OF $\lambda(l)$ AND $\lambda(l, j)$

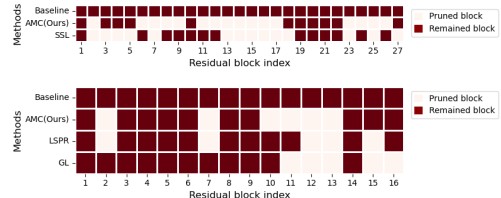

Figure 3: The compressed ResNet56 on CIFAR-10 and ResNet34 on ImageNet.

Table 2: Performance comparisons of SSL (Wen et al., 2016), commonly used group lasso (GL) and AMSC for ResNet56 on CIFAR-10 and ResNet34 on ImageNet.

| Datasets | Methods | Acc.(%) | FLOPs(M/B)($\downarrow$) | Params.(M)($\downarrow$) | Depth |
|---|---|---|---|---|---|
| CIFAR-10 | Baselines | 93.37 | 125.48(-) | 0.85(-) | 56 |
| | SSL | 93.25 | 59.79(52.92%) | 0.50(41.18%) | 28 |
| | AMSC(Ours) | **93.71** | **51.91(58.63%)** | **0.47(44.71%)** | **24** |
| ImageNet | Baselines | 73.31 | 3.66(-) | 21.80(-) | 34 |
| | GL | 72.15 | 2.52(31.34%) | **8.82(59.54%)** | 24 |
| | AMSC(Ours) | **72.93** | **2.27(38.15%)** | 16.61(23.81%) | **22** |

Here, we investigate the the impact of the choice of $\lambda(l)$ and $\lambda(l, j)$ in equation 3. For visualization, we only demonstrate the impact of the choice of $\lambda(l)$ by comparing the proposed $\lambda(l) = \frac{1}{\|\hat{\theta}_l\|_2} \sqrt{q_l}$ with the setting $\lambda(l) = 1$ in SSL (Wen et al., 2016) and $\lambda(l) = \sqrt{q_l}$ in a group lasso setting (GL) (Yuan & Lin, 2006; WEI & HUANG, 2010). As shown in Table 2, the proposed AMSC surpasses SSL in accuracy and all compression metrics, while it also outperforms GL in accuracy and FLOPs, although not in parameter counts. To see more clearly, we show the compressed networks in Figure 3. Obviously, SSL treats layers with varying parameter counts equally, leading to a tendency to compress layers with fewer parameters (Top in Figure 3). Conversely, GL heavily weights layers with larger parameters, leading to significant compression of these layers (Bottom in Figure 3). Hence, neither setting achieves precise and efficient compression. In contrast, AMSC adaptively adjusts the penalty for each layer based on its importance and parameter counts, preserving the earlier layers while compressing the middle layers more extensively. This aligns with the current understanding of neural networks. Particularly, it is well known that the earlier layers usually extract features such as edges, texture and color, which serve as fundamental components for later layers and should be preserved. Conversely, the outputs of the middle layers often show similar features (Nguyen et al., 2020) and should be compressed.

By setting an appropriate $\lambda(l)$, AMSC effectively distinguishes critical and redundant layers, and achieves a more precise and effective compression.

### 5.3.4 MULTI-DIMENSIONAL COMPRESSION V.S. SINGLE-DIMENSIONAL COMPRESSION

In this section, we investigate the advantages of multi-dimensional compression compared to single-dimensional compression. We implement width-only and depth-only compressions by setting $\lambda_0 = 0$ and $\lambda_1 = 0$ in (3), respectively. Figure 4 illustrates that the accuracy of all three methods initially increases and then decreases as FLOPs or parameter counts are reduced. Notably, under similar FLOPs (parameter counts), AMSC consistently achieves higher accuracy across all architectures and datasets. This suggests that multi-dimensional compression is more effective at identifying reasonable substructures compared to single-dimensional compression under a given computational budget.

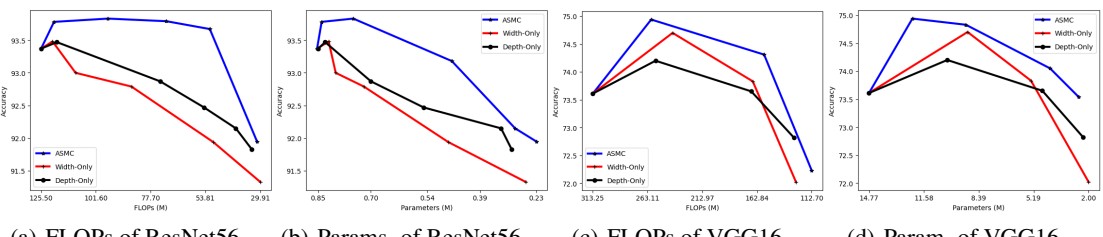

(a) FLOPs of ResNet56    (b) Params. of ResNet56    (c) FLOPs of VGG16    (d) Param. of VGG16

Figure 4: The accuracy varies with FLOPs (Parameter counts) reducing.

### 5.3.5 TRAINING BUDGETS AND INFERENCE TIME

Table 3: The training complexity comparison for ResNet56 on CIFAR-10.

| Methods | DBP | LPSR | ELC | AMSC |
|---------|-----|------|-----|------|
| Epochs  | 320 | 320  | 600 | 500  |

Table 4: The average inference time comparison for ResNet56 on CIFAR-10 using an NVIDIA A100 GPU with a batch size of 1 (100 Trials).

| Methods | W | D | Acc.↑(%) | FLOPs | Inference time | Speedup Ratio |
|---------|---|---|----------|-------|----------------|---------------|
| Baseline |   |   | 0.00 | 125.48M | 4.22ms | 0.00 |
| HRank | ✓ |   | -0.09 | 62.70M | 2.34ms | 1.80 |
| FPGM | ✓ |   | -0.33 | 59.40M | 2.63ms | 1.60 |
| ELC |   | ✓ | 0.21 | 58.30M | 2.01ms | 2.10 |
| AMSC | ✓ | ✓ | **0.34** | **51.91M** | **1.91ms** | **2.21** |

Compared to traditional pruning-based compression methods, which repeatedly fine-tune to offset performance degradation caused by pruning, AMSC does not incur high training costs. We present the training complexities of several existing state-of-the-art methods (Wang et al., 2019; Xu et al., 2022; Wu et al., 2023) for ResNet56 on CIFAR-10 in Table 3. As we can see, our training complexity is comparable to these methods, and in some cases, it may even be lower.

We further evaluate the average inference times of models compressed by different methods for ResNet56 on CIFAR-10, conducted on a NVIDIA A100 GPU with a batch size of 1, and repeat the tests 100 times. As shown in Table 4, the resulting model of AMSC delivers faster inference speeds due to its minimal structures.

## 6 CONCLUSION

In this paper, we introduce an adaptive multi-dimensional structured compression (AMSC) method to reduce both depth and width of the networks. To adaptively identify the redundant structures, we apply the weighted adaptive group penalty to the parameters of each components. Our approach is supported by rigorous theoretical evidence demonstrating its consistency in achieving minimal network structure. Extensive experiments conducted on CIFAR-10/100 and ImageNet datasets demonstrate that our proposed AMSC method not only achieves state-of-the-art compression performance measured by FLOPs and parameter counts but also maintains competitive classification performance. We will expand the proposed AMSC to more visual tasks such as object detection and image segmentation in the future works.

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

# A PROOF OF MAIN THEOREMS IN SECTION 4

## A.1 PROOF OF THEOREM 4.1

Recall that the optimal parameters set $\Theta^*(w) = \{\theta^* : \theta^* \in \arg\min_\theta \mathbb{E}_{(X,y)\sim\mu}\mathcal{L}_{target}(y, f(X, \theta)\}$. Before giving the complete proof of Theorem 4.1, we assume the following assumptions and restate Theorem 4.1.

**Assumption 4.1.** *For any $\theta_1^*, \theta_2^* \in \Theta^*$, $Q_d(\theta_1^*) \leq Q_d(\theta_2^*)$ implies $dep(\theta_1^*) \leq dep(\theta_2^*)$ and $Q_w(\theta_1^*) \leq Q_w(\theta_2^*)$ implies $wid(\theta_1^*) \leq wid(\theta_2^*)$.*

**Assumption 4.2.** *The loss function $\mathcal{L}_{target}(\theta)$ is a sub-analytic function related to $\theta$.*

**Assumption 4.3.** *For any $\theta_1^*, \theta_2^* \in \Theta^*$, $\|\theta_1^* - \theta_2^*\|_2 \leq M_b < +\infty$.*

**Theorem 4.1.** *Suppose that Assumptions 4.1, 4.2 and 4.3 hold. Let $\hat{\theta}$ be the estimator of equation 3, if $\lambda_0 = o_p(1)$, $\lambda_1 = o_p(\lambda_0)$ and the statistical error $S_n = o_p(\lambda_1)$, we deduce that*

$$\mathbb{P}(dep(\hat{\theta}) = l_\theta) \to 1, \ \mathbb{P}(wid(\hat{\theta}) = w_\theta(l_\theta)) \to 1, \ d(\hat{\theta}, \Theta^*)\hat{=}\min_{\theta^*\in\Theta^*}\|\hat{\theta} - \theta^*\|_2 = o_p(1). \tag{6}$$

*Proof.* For simplicity, we denote $\hat{\mathcal{L}}(\theta) = \frac{1}{n}\sum_{i=1}^n \mathcal{L}_{target}(y_i, f(X_i, \theta))$ and $\mathcal{L}(\theta) = \mathbb{E}_{(X,y)\sim\mu}\mathcal{L}_{target}(y, f(X, \theta))$. Due to the definition of $\hat{\theta}$, for any $\theta^* \in \Theta^*$, we have

$$\hat{\mathcal{L}}(\hat{\theta}) + \lambda_0 Q_d(\hat{\theta}) + \lambda_1 Q_w(\hat{\theta}) \leq \hat{\mathcal{L}}(\theta^*) + \lambda_0 Q_d(\theta^*) + \lambda_1 Q_w(\theta^*)$$
$$\iff \hat{\mathcal{L}}(\hat{\theta}) - \hat{\mathcal{L}}(\theta^*) + \lambda_0(Q_d(\hat{\theta}) - Q_d(\theta^*)) + \lambda_1(Q_w(\hat{\theta}) - Q_w(\theta^*)) \leq 0. \tag{7}$$

Since $Q_d(\theta)$ and $Q_w(\theta)$ are both Lipschitz functions, then there exists a constant $c_2$ such that

$$\mathcal{L}(\hat{\theta}) - \mathcal{L}(\theta^*) \leq \mathcal{L}(\hat{\theta}) - \hat{\mathcal{L}}(\hat{\theta}) - \mathcal{L}(\theta^*) + \hat{\mathcal{L}}(\theta^*) + \lambda_0(Q_d(\theta^*) - Q_d(\hat{\theta})) + \lambda_1(Q_w(\theta^*) - Q_w(\hat{\theta}))$$
$$\leq |\mathcal{L}(\hat{\theta}) - \hat{\mathcal{L}}(\hat{\theta})| + |\mathcal{L}(\theta^*) - \hat{\mathcal{L}}(\theta^*)| + \lambda_0(Q_d(\theta^*) - Q_d(\hat{\theta})) + \lambda_1(Q_w(\theta^*) - Q_w(\hat{\theta})) \tag{8}$$
$$\leq \mathcal{O}_p(S_n) + (\lambda_0 + \lambda_1)c_2\|\theta^* - \hat{\theta}\|_2.$$

The first inequality arises from the fact that the term is smaller than its sum with a positive value and $S_n$ is the statistic error (Jiao et al., 2023; Dinh & Ho, 2020) of any $\theta \in \Theta$ in the third inequality. Mathematically, denote the class of neural network is $\mathcal{F}$. The statistical error $S_n$ can be bounded by the pseudo dimension of $\mathcal{F}$, denoted by Pdim($\mathcal{F}$). In particular, if both the architecture and activation functions within $\mathcal{F}$ remain fixed, it follows that Pdim($\mathcal{F}$) = VCdim($\mathcal{F}$) (Bartlett, 1996), where VCdim($\mathcal{F}$) is the Vapnik-Chervonenkis (VC) dimension of $\mathcal{F}$, which can be further bounded by width, depth and the number of parameters of $\mathcal{F}$ (Jiao et al., 2023; Bartlett et al., 2019). There are several existing results on the statistical error. Chen et al. (2020a) demonstrate that it scales as $\mathcal{O}_p(n^{-\frac{\beta}{2\beta+d}})$ where $\beta$ is the smoothness index of true function class and $d$ is the input dimension. This rate can be further refined to $\mathcal{O}_p(n^{-\frac{\beta}{2\beta+d^*}})$, where $d^*$ is the intrinsic dimension of data (Nakada & Imaizumi, 2020).

On the one hand, $\Theta^*$ is the zero set of $\mathcal{L}(\hat{\theta}) - \mathcal{L}(\theta^*)$. Denote $\eta = \arg\min_{\theta^*\in\Theta^*}\|\theta^* - \hat{\theta}\|_2$. Hence, by Lemma A.1 and Assumption 4.2, there exists constants $c_1 > 0$ and $v > 2$ such that

$$c_1 d(\hat{\theta}, \Theta^*)^v = c_1\|\eta - \hat{\theta}\|_2^v \leq \mathcal{L}(\hat{\theta}) - \mathcal{L}(\eta) \leq \mathcal{O}_p(S_n) + (\lambda_0 + \lambda_1)c_2\|\eta - \hat{\theta}\|_2. \tag{9}$$

On the other hand, Using the Young's inequality, there exists a constant $c_1$ such that

$$c_2(\lambda_0 + \lambda_1)\|\eta - \hat{\theta}\|_2 \leq \frac{1}{v}\left(\frac{(c_1 v)^{1/v}}{2}\|\eta - \hat{\theta}\|_2\right)^v + \frac{v-1}{v}\left(\frac{2c_2}{(c_1 v)^{1/v}}(\lambda_0 + \lambda_1)\right)^{v/(v-1)}$$
$$= \frac{c_1}{2}\|\eta - \hat{\theta}\|_2^v + \frac{(v-1)(2c_2)^{v/(v-1)}}{v(c_1 v)^{1/(v-1)}}(\lambda_0 + \lambda_1)^{v/(v-1)}. \tag{10}$$

Combining the two inequality, we have

$$\frac{c_1}{2}\|\eta - \hat{\theta}\|_2^v \leq \mathcal{O}_p(S_n) + \frac{(v-1)(2c_2)^{v/(v-1)}}{v(c_1 v)^{1/(v-1)}}(\lambda_0 + \lambda_1)^{v/(v-1)}. \tag{11}$$

Collate the above results, we have

$$d(\hat{\theta}, \Theta^*) = \mathcal{O}_p((S_n + (\lambda_0 + \lambda_1)^{v/(v-1)})^{1/v}). \tag{12}$$

equation 12 indicates that if $\lambda_0, \lambda_1 = o_p(1)$, $d(\hat{\theta}, \Theta^*)$ will vanishes with probability going to 1, i.e. $d(\hat{\theta}, \Theta^*) = o_p(1)$ and $\hat{\theta} \to \Theta^*$.

Furthermore, by equation 7, for any $\theta^* \in \Theta^*$, we have

$$\hat{\mathcal{L}}(\hat{\theta}) + \lambda_0 Q_d(\hat{\theta}) + \lambda_1 Q_w(\hat{\theta}) \leq \hat{\mathcal{L}}(\theta^*) + \lambda_0 Q_d(\theta^*) + \lambda_1 Q_w(\hat{\theta})$$
$$\iff \lambda_0 Q_d(\hat{\theta}) \leq \lambda_0 Q_d(\theta^*) + \lambda_1(Q_w(\hat{\theta}) - Q_w(\theta^*)) + \hat{\mathcal{L}}(\theta^*) - \mathcal{L}(\theta^*) - \hat{\mathcal{L}}(\hat{\theta}) + \mathcal{L}(\hat{\theta}) + \mathcal{L}(\theta^*) - \mathcal{L}(\hat{\theta})$$
$$\iff Q(\hat{\theta}) \leq Q(\theta^*) + \frac{\lambda_1}{\lambda_0}\|\hat{\theta} - \theta^*\|_2 + \mathcal{O}_p(\frac{1}{\lambda_0}S_n)$$
$$\iff Q(\hat{\theta}) \leq Q(\theta^*) + \frac{\lambda_1}{\lambda_0}d(\hat{\theta}, \Theta^*) + \frac{\lambda_1}{\lambda_0}M_b + \mathcal{O}_p(\frac{1}{\lambda_0}S_n). \tag{13}$$

The third inequality arises from that $Q_w(\theta)$ is a Lipschitz function and the final inequality is because of triangle inequality and Assumption 4.3. Consequently, in equation 13, if $\lambda_1 = o(\lambda_0)$ and $S_n = o_p(\lambda_0)$, we deduce that $Q(\hat{\theta}) \leq Q(\theta^*)$ as $n \to \infty$. Due to the arbitrariness of $\theta^*$ and Assumption 4.1, we have $\mathbb{P}(\mathrm{dep}(\hat{\theta}) = l_\theta) \to 1$ as $n \to \infty$.

Finally, to prove that $\hat{\theta}$ can achieve the minimal width, we denote $\bar{\Theta}^* = \{\gamma^* : \gamma^* \in \Theta^*, \mathrm{dep}(\gamma^*) = l_\theta\}$, which is the optimal parameter set with minimal depth. Then, we can construct an $\bar{\theta}^* \in \bar{\Theta}^*$ such that $Q_d(\bar{\theta}) = Q_d(\hat{\theta})$ and $\mathrm{wid}(\bar{\theta}) = w_\theta(l_\theta)$. This construction always exists due to the unidentifiability of neural network. Therefore, by equation equation 7 again, we have

$$\hat{\mathcal{L}}(\hat{\theta}) + \lambda_0 Q_d(\hat{\theta}) + \lambda_1 Q_w(\hat{\theta}) \leq \hat{\mathcal{L}}(\bar{\theta}^*) + \lambda_0 Q_d(\bar{\theta}^*) + \lambda_1 Q_w(\bar{\theta}^*)$$
$$\iff \hat{\mathcal{L}}(\hat{\theta}) + \lambda_1 Q_w(\hat{\theta}) \leq \hat{\mathcal{L}}(\bar{\theta}^*) + \lambda_1 Q_w(\bar{\theta}^*) \tag{14}$$
$$\iff Q_w(\hat{\theta}) \leq Q_w(\bar{\theta}^*) + \mathcal{O}_p(\frac{1}{\lambda_1}S_n)$$

Hence, if $S_n = o_p(\lambda_1)$, we deduce that $Q_w(\hat{\theta}) \leq Q_w(\bar{\theta})$ as $n \to \infty$. By Assumption 4.1, we have $\mathbb{P}(\mathrm{wid}(\hat{\theta}) = w_\theta(l_\theta)) \to 1$ as $n \to \infty$.

$\square$

## A.2 RELATED LEMMAS

To provide a polynomial convergence rate of $\hat{\theta}$, we need the following Lemma.

**Lemma A.1.** *There exist $c_1 > 0$ and $v > 2$ such that $\mathcal{L}(\theta) - \mathcal{L}(\theta^*) \geq c_1 d(\theta, \Theta^*)^v$ for all $\theta \in \Theta$ and $\theta^* \in \Theta^*$.*

*Proof.* Firstly, since $\mathcal{L}(\theta)$ is sub-analytic related to $\theta$ by Assumption 4.2, the excess risk $g_{\mathcal{L}}(\theta) = \mathcal{L}(\theta) - \mathcal{L}(\theta^*)$ is also sub-analytic in $\theta$. Thus $\Theta^*$ is the zero level-set of the sub-analytic function $g_{\mathcal{L}}(\theta)$. By Lojasiewicz inequality (Ji et al., 1992; Colding & Minicozzi, 2014; Bolte et al., 2006) for algebraic varieties, there exists positive constants $c'_1 > 0$ and $v > 2$ such that $d(\theta, \Theta^*)^v \leq c'_1 |g_{\mathcal{L}}(\theta)| \ \forall \theta \in \Theta$, which completes the proof. □

# B  DETAILED IMPLEMENT DETAILS AND MORE RESULTS

## B.1  ALGORITHMS

We implement the proposed AMSC by Algorithm 1. Since the parameters trained with group lasso may not converge exactly to zero, we use a threshold to preserve the important layers and filters after training. Additionally, the group lasso penalty may shrink the remaining parameters (Tibshirani, 1996; Zou, 2006), potentially compromising performance. Therefore, we train the pruned model with slight extra budgets by continue optimizing equation 3 without penalty terms.

---
**Algorithm 1:** Adaptive Multi-dimensional Structured Compression

---
**Require:** A baseline model $\hat{\theta}$, penalty parameters $\lambda_0$ and $\lambda_1$, pruning thresholds $\tau_0$ and $\tau_1$;
**Output:** A compressed model;
 1: Compute $\lambda(l)$ and $\lambda(l, j)$ based on $\hat{\theta}$ by equation 4;
 2: **for** number of training iterations **do**
 3:     Compute the loss by equation 3;
 4:     Update parameters;
 5: **end for**
 6: Prune the depth units whose $L_2$ norms are smaller than $\tau_0$;
 7: Prune the width units whose $L_2$ norms are smaller than $\tau_1$;
 8: **for** number of penalty-free training iterations **do**
 9:     Compute the loss by equation 3 without penalty terms ($\lambda_0 = 0$, $\lambda_1 = 0$);
10:     Update the remained parameters;
11: **end for**
12: **return** compressed model.

---

## B.2  THE ARCHITECTURE MODIFICATION OF VGGS

To maintaining the network connectivity when compressing the depth of VGGs, we modify the convolution operation as $z_l = ReLU(w_l * z_{l-1}) = ReLU(\tilde{w}_l * z_{l-1} + z_{l-1})$, where $w_l$, $\tilde{w}_l$ are the learned $3 \times 3$ convolutional kernels and $*$ is the convolution operator. As the convolution operation is linear, This reformulation does not compromise the expressive capacity of the convolutional layers. Consequently, given $ReLU(ReLU(x)) = ReLU(x)$, if $\tilde{w}_l = 0$, we deduce

$$z_l = ReLU(z_{l-1}) = ReLU(ReLU(w_{l-2} * z_{l-2}) = ReLU(w_{l-2} * z_{l-2}) = z_{l-1}.$$

### B.3 EXPERIMENTAL SETTINGS

For publicly available pretrained models, such as DeiT series, we directly use them as the baselines. Otherwise, we train all networks from scratch to establish baseline models. [1] In the implementation of AMSC, following Chen & Zhao (2019) and Wang et al. (2019), the depth penalty is only enforced on the layers (blocks) between each pair of shortcut endpoints, excluding the first convolutional layer. Unless otherwise stated, the optimization algorithm, initial network values, learning rates, and schedules remain consistent between the proposed method and the baseline. All computations are performed using PyTorch (Paszke et al., 2019) in Python. The detailed training settings are as follows:

**Training settings**: For base models trained on CIFAR-10, we set batch size to 64 for DenseNet40 and 128 for ResNet56/110, respectively. Weight decay is set to $10^{-4}$. The DenseNets are trained for 160 epochs with the learning rate starting from 0.1 and divided by 10 at epochs 80 and 120. And the ResNets are trained for 200 epochs with the learning rate starting from 0.1 and divided by 10 at epochs 100 and 150. These are all the the most training settings (He et al., 2016; Huang et al., 2017) for models trained on CIFAR-10. On CIFAR-100, the training settings for ResNet56 are the same as that in CIFAR-10. For VGG16/19, the batch size is 128 and the weight decay is $10^{-4}$. The VGGs are trained for 200 epochs with the learning rate starting from 0.1 and divided by 5 at epochs 60, 120 and 160. These are also common used settings (Liu et al., 2017; Lin et al., 2020). On ImageNet, following (He et al., 2016), we train ResNet34 for 90 epochs with batch size is 256 and the learning rate starting from 0.1 and divided by 10 at epochs 30 and 60. For DeiT, we follow the standard training configurations as stated in Touvron et al. (2021). The pruning threshold are set as follows. For all CNNs experiments, we set depth pruning threshold $\tau_0 = 0.5$ and width pruning threshold $\tau_1 = 0.01$, while for DeiT, $\tau_0 = 1.0$ and $\tau_1 = 0.01$. Finally, in the extra penalty-free training phase, we apply a linear learning restarting strategy (Zimmer et al., 2022). Specifically, the initial learning rate is 0.1 for all models on CIFAR-10/100 with 100 epochs and 0.01 for ResNet34 on ImageNet with 240 epochs. In addition, the DeiT experiments do not involve extra training budgets, following Lin et al. (2024).

**The selection strategy for penalty parameters and their settings.** In AMSC, $\lambda_0$ controls depth and $\lambda_1$ controls width. Following our depth-first strategy, these parameters are selected sequentially. Specifically, $\lambda_0$ is first selected by minimizing predictive error in the validation dataset without considering width compression ($\lambda_1 = 0$). Then, keeping $\lambda_0$ fixed at the selected value, $\lambda_1$ is chosen to specify the width. On CIFAR-10/100, $\lambda_0$ is chosen in $\{0.1, 0.5, 0.8, 1, 5, 8\}$, $\lambda_1$ is chosen in $\{0.0001, 0.0005, 0.0008, 0.001, 0.002, 0.003\}$. On ImageNet, for ResNet34 and ResNet50, $\lambda_0$ is chosen in $\{0.05, 0.06, 0.07, 0.08, 0.09\}$ and $\lambda_1$ is chosen in $\{0.00001, 0.00002, 0.00003\}$. For DeiT, both $\lambda_0$ and $\lambda_1$ are chosen from $\{0.00001, 0.00002, 0.00003\}$.

### B.4 MORE RESULTS ON CIFAR-10/100

Experimental results on CIFAR-10 for VGG16, and CIFAR-100 for ResNet56, VGG16, and VGG19 are summarized in Table 5. For CIFAR-10, AMSC achieves a more sufficient compression results with performance guarantee. For CIFAR-100, across all three architectures, AMSC consistently achieves significantly fewer FLOPs and reduced parameter counts when compared to existing methods, while maintaining a relatively similar accuracy reduction. For ResNet56, although DLRFC (He et al., 2022) achieves a higher accuracy decrement, AMSC delivers a substantial decrease in FLOPs (58.92M vs. 92.87M) accompanied by a comparable decrease in accuracy (-0.44% vs. 0.27%). Similar patterns are observed for VGG16, where AMSC and APIB (Guo et al., 2023a) present analogous outcomes. In the case of VGG19, methods such as

---

[1]The baseline models used in this study are directly implemented from the following GitHub repositories:
ResNet56/110: https://github.com/akamaster/pytorch_resnet_cifar10;
ResNet34: https://github.com/pytorch/examples/tree/main/imagenet;
DenseNet: https://github.com/andreasveit/densenet-pytorch;
VGG: https://github.com/weiaicunzai/pytorch-cifar100/tree/master;
DeiT: https://github.com/facebookresearch/deit.

Table 5: Performance comparisons for ResNet56, VGG16 and VGG19 on CIFAR-100. Pruned and Acc.↑ denote pruned accuracy and relative accuracy increase, respectively. The positive values in Acc.↑ are colored by blue and the best scores in each block are highlighted via bold text.

| Dataset | Architecture | Methods | W | D | Baseline(%) | Pruned(%) | Acc↑(%) | FLOPs(M) | Params.(M) |
|---------|--------------|---------|---|---|-------------|-----------|---------|----------|------------|
| CIFAR-10 | VGG16 | SDN(Chen & Zhao, 2019) | | ✓ | 93.50 | 93.47 | -0.03 | 191.38 | **1.79** |
| | | GAL(Lin et al., 2019b) | ✓ | ✓ | 93.96 | 92.03 | -1.93 | 189.49 | 3.36 |
| | | HRank(Lin et al., 2020) | ✓ | | 93.96 | 93.43 | -0.53 | 145.61 | 2.51 |
| | | ELC(Wu et al., 2023) | | ✓ | 93.56 | 93.25 | -0.31 | 144.90 | - |
| | | APIB(Guo et al., 2023a) | ✓ | | 93.68 | 94.08 | **0.40** | 125.30 | 3.55 |
| | | CPMC(Yan et al., 2021) | ✓ | | 93.68 | 93.40 | -0.28 | 106.50 | - |
| | | AMSC(Ours) | ✓ | ✓ | 93.60 | 93.46 | -0.14 | **100.98** | 1.96 |
| CIFAR-100 | ResNet56 | DLRFC(He et al., 2022) | ✓ | | 71.14 | 71.41 | **0.27** | 92.87 | - |
| | | SDN(Chen & Zhao, 2019) | | ✓ | 70.01 | 69.78 | -0.23 | 78.40 | 0.55 |
| | | LPSR(Zhang & Liu, 2022) | | ✓ | 71.39 | 70.17 | -1.22 | 60.20 | 0.54 |
| | | PGMPF(Cai et al., 2022) | ✓ | | 72.92 | 70.21 | -2.71 | 58.98 | - |
| | | APIB(Guo et al., 2023a) | ✓ | | 72.52 | 70.89 | -1.63 | 58.98 | - |
| | | AMSC(Ours) | ✓ | ✓ | 70.05 | 69.61 | -0.44 | **58.92** | **0.49** |
| | VGG-16 | SDN(Chen & Zhao, 2019) | | ✓ | 72.38 | 73.43 | **1.05** | 210.18 | 3.12 |
| | | CPGMI(Lee et al., 2020) | ✓ | | 73.80 | 73.53 | -0.27 | 197.34 | - |
| | | CPMC(Yan et al., 2021) | ✓ | | 73.80 | 73.01 | -0.79 | 162.88 | - |
| | | PGMPF(Cai et al., 2022) | ✓ | | 73.80 | 73.66 | -0.14 | 162.88 | - |
| | | APIB(Guo et al., 2023a) | ✓ | | 73.80 | 73.89 | 0.09 | 162.88 | - |
| | | AMSC(Ours) | ✓ | ✓ | 73.61 | 73.54 | -0.07 | **148.20** | 2.58 |
| | VGG-19 | Slimming(Liu et al., 2017) | ✓ | | 73.26 | 70.92 | -2.34 | 127.00 | - |
| | | ELC(Wu et al., 2023) | | ✓ | 71.28 | 70.03 | -1.25 | **124.50** | - |
| | | AMSC(Ours) | ✓ | ✓ | 72.18 | 71.73 | **-0.45** | 133.94 | **2.34** |

Slimming (Liu et al., 2017) and ELC (Wu et al., 2023) sacrifice accuracy to compress the network. In stark contrast, AMSC achieves comparable reductions in FLOPs with negligible loss in accuracy. These results underscore AMSC's capability to preserve accuracy while achieving a minimal architecture. Furthermore, the architectures optimized via AMSC for both VGG16 and VGG19 demonstrate comparable accuracy and FLOPs. This consistency again shows the existence of a minimal architecture that is robust across various initial architectures and the ability of AMSC to achieve it.

## B.5 MORE EMPIRICAL VERIFICATION OF ASSUMPTION 4.1

We verify the monotonic relationship between $dep(\theta)$ and $Q_d(\theta)$ for any $\theta^* \in \Theta^*$ in Figure 5 based on six ResNets on CIFAR-10.

We verify the monotonic relationship between $dep(\theta)$ and $Q_d(\theta)$ for any $\theta^* \in \Theta^*$ in various width in Figure 6 based on ResNet56 on CIFAR-10.

## B.6 MORE RESULTS ABOUT THE INFLUENCE OF $\lambda_0$ AND $\lambda_1$

We further demonstrate the influence of $\lambda_0$ in Figure 7 based on various ResNets on CIFAR-10. All results show a similar pattern.

## B.7 INFLUENCE OF PRUNING THRESHOLDS

We investigate the influence of pruning thresholds in this section. Because of the adaptive weight in penalty terms where the weights for unimportant components are considerably large, the $L_2$ norm of the unimportant components are almost zero when using AMSC. As shown in Table 6, the $L_2$ norm of pruned layers (filters) is markedly smaller than that of remained layers (filters). Hence, the threshold can be easily set and the final network structure is quite robust to the threshold.

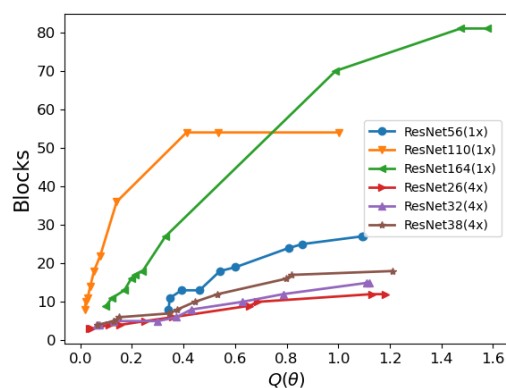

Figure 5: Relationship between $\text{dep}(\theta)$ and $Q_d(\theta)$ for any $\theta^* \in \Theta^*$ in different architectures. The accuracy of each point meets or exceeds the baseline and varies by less than $1\%$ across different points within the same architecture. Therefore, these points can be approximately considered as elements of $\Theta^*$.

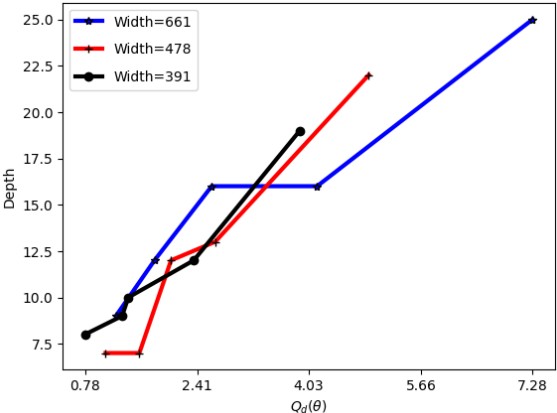

Figure 6: Relationship between $\text{dep}(\theta)$ and $Q_d(\theta)$ for any $\theta^* \in \Theta^*$ for ResNet56 on different width. The accuracy of each point meets or exceeds the baseline and varies by less than $1\%$ across different points within the same architecture. Therefore, these points can be approximately considered as elements of $\Theta^*$.

Table 6: The $L_2$ norm of pruned and remained layers (filters)

| Datasets | Architecture | Pruned layers | Remained layers | Pruned filters | Remained filters |
|---|---|---|---|---|---|
| CIFAR-10 | ResNet56 | 0.012 | 4.578 | 0.001 | 0.811 |
| ImageNet | ResNet34 | 0.030 | 12.196 | 0.002 | 0.881 |
| ImageNet | DeiT-tiny | 0.091 | 11.971 | 0.005 | 0.433 |

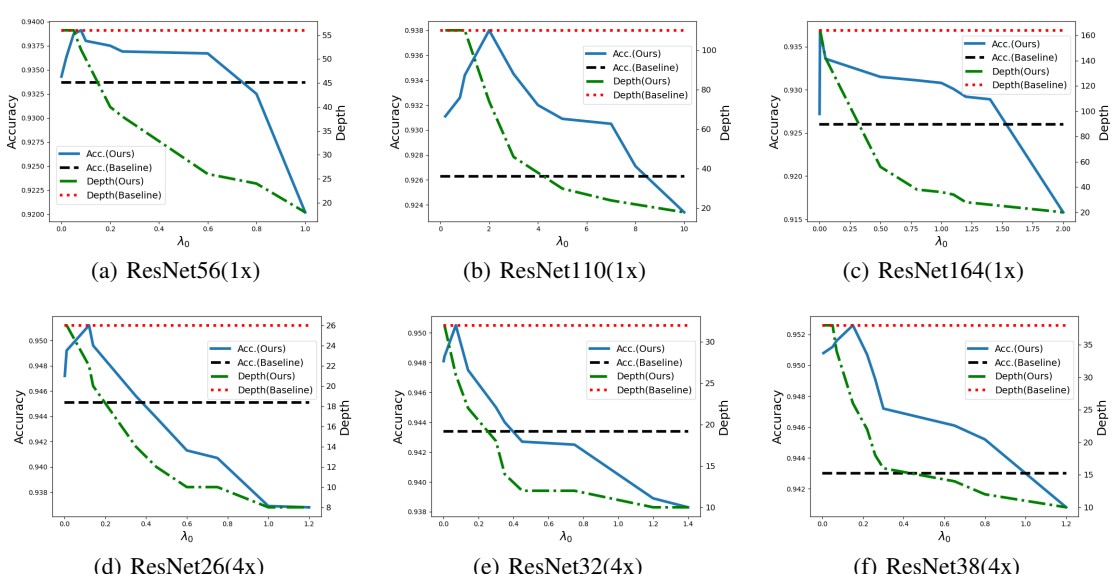

Figure 7: The accuracy and depth of different ResNets vary with the penalty parameter $\lambda_0$ on CIFAR-10. In each figure, the left Y-axis denotes the accuracy and the right Y-axis denotes the depth.