# OpenReview forum: "AMSC: Adaptive Multi-Dimensional Structured Compression with Theoretical Guarantees"
_ICLR.cc/2025/Conference — Submitted to ICLR 2025_

### Official Review · Reviewer_wizb · 2024-11-01

**Soundness:** 3
**Presentation:** 4
**Contribution:** 2
**Rating:** 6
**Confidence:** 4

**Summary:**

This paper proposes an adaptive multi-dimensional structured pruning method to compress DNNs by incorporating the penalty terms related to filters and layers into the loss function. Additionally, the adaptive weights are leveraged to modulate the influence of the penalty for each filter in each individual layer. Experiments on classification tasks and theoretical evidence demonstrate that the proposed method seems to be effective.

**Strengths:**

1. Code is released.
2. The paper is generally well-written, and the theoretical analysis is comprehensive.
3. The experiments are extensive, with various datasets and models.

**Weaknesses:**

1. In Table 1, the baselines in this paper are significantly lower than those in previous methods. For example, for ResNet-110, the baselines of other methods are all above 93.5%, while the baseline in this paper is 92.63% (a reduction of >0.87%). The authors should compare the results among different methods under similar baselines.
2. The authors should report the results of VGG-16 on CIFAR-10 and ResNet-50 on ImageNet, as numerous methods conduct experiments on these architectures, such as [1, 2, 3, 4, 5]. Reporting these results would facilitate comparisons with other approaches.\
[1] M. Lin, R. Ji, Y. Wang, et al. Hrank: Filter pruning using high-rank feature map. CVPR 2020.\
[2] H. Zhang, L. Liu, H. Zhou, et al. Akecp: Adaptive knowledge extraction from feature maps for fast and efficient channel pruning. ACMMM 2021.\
[3] H. Wang, C. Qin, Y. Zhang, et al. Neural pruning via growing regularization. ICLR 2021.\
[4] G. Fang, X. Ma, M. Song, et al. Depgraph: Towards any structural pruning. CVPR 2023.\
[5] L. Liu, S. Zhang, Z. Kuang, et al. Group fisher pruning for practical network compression. ICML 2021.
3. To demonstrate the effectiveness of Equation 4, more experiments should be conducted in Section 5.3.3, including: \
(a) Keep $ \lambda _0 $, $ \lambda _1 $  unchanged, compare the results of proposed method with those obtained when setting $\lambda (l) = 1$, $\lambda (l,j) = 1$.\
(b) Compare the results with those obtained using reverse weighting, i.e., when $\frac{{\sqrt {q _ l  } }}{{\lVert {\hat \theta _ l  } \lVert_2 }}$ is larger, the corresponding weight $\lambda (l)$ should be smaller. Similarly, when $ \frac{{\sqrt {q _ {l,j} } }}{{\lVert {\hat \theta _ {l,j} } \lVert_2 }} $ is larger, the corresponding weight $\lambda (l,j)$ should be smaller.\
(c) Analyze the influence of $\lambda (l)$ and $\lambda (l,j)$ on accuracy individually, as well as their influence when investigating $\lambda (l)$ and $\lambda (l,j)$ simultaneously.
4. Do the authors validate the effectiveness of the proposed method on other tasks, such as object detection?

\
Minor issues:
1. Several symbol errors. For example, $ \lVert {\theta _ l } \lVert \propto \sqrt {q _ l } $ and $\lVert \theta _ {l,j} \lVert \propto \sqrt{q _ {l,j}}$ seem to lack rigor. Also, there is an error in the equation on page 5, Line 209.

**Questions:**

Please see the weaknesses.

---

> ### Author Response · Authors · 2024-11-25
> **Part1: The response to weaknesses 1.**
>
> Thank you for your insightful comments and suggestions, which have greatly improved the manuscript.
>
> **W1:** In Table 1, the baselines in this paper are significantly lower than those in previous methods. For example, for ResNet-110, the baselines of other methods are all above 93.5\%, while the baseline in this paper is 92.63\% (a reduction of >0.87\%). The authors should compare the results among different methods under similar baselines.
>
> **A1:** The baseline implementations for ResNet56 and ResNet110 on CIFAR-10 are sourced directly from the GitHub repository: https://github.com/akamaster/pytorch_resnet_cifar10. The previous experiments are conducted using PyTorch 1.12.1 on an NVIDIA A40 GPU where the reproduced results are slightly lower than  those reported in the repository. However, when tested in a new experimental environment (PyTorch 2.4.0 with an NVIDIA GeForce RTX 4090 GPU), we find that it could achieve results similar to those reported in other papers. **This indicates that the fluctuation of baseline results might caused by the variation of experimental environments.**
>
> We re-implement the proposed AMSC method on a new baseline for evaluation, which is completely based on the code attached in the supplementary. As shown in the following Table 1, the updated baselines for ResNet56 and ResNet110 on CIFAR-10 achieve accuracies of 93.37\% and 93.51\%, respectively. Using AMSC, these models attain accuracies of 93.71\% and 92.73\%, while simultaneously reducing computational cost by 58.63\% and 78.37\% and parameters by 44.70\% and 80.34\%, respectively, establishing new state-of-the-art (SOTA) results for these architectures.
>
> **Table 1: Performance comparisons for various architectures on CIFAR-10.** Pruned and Acc.$\uparrow$ denote pruned accuracy and relative accuracy increase, respectively. W and D indicate whether the model will be pruned along depth and width, respectively.The highest scores in each block are highlighted in bold, while the second highest scores are italics.
> | Dataset    | Architecture | Methods                        | W          | D          | Baseline (%) | Pruned (%) | Acc$\uparrow$(%)  | FLOPs (M/B) | Params. (M) |
> |:----------:|:------------:|:------------------------------:|:----------------:|:----------------:|:---------------------:|:-------------------:|:---------------------:|:-----------------:|:-----------------:|
> | CIFAR-10   | ResNet56     | GAL [8]           | ✅          | ✅          | 93.26        | 93.38      | 0.12       | 78.74       | 0.75        |
> |            |              | DLP [6] |            | ✅          | -            | -          | -0.82      | 65.80       | 0.52        |
> |            |              | TDPF [11]      | ✅          | ✅          | 93.69        | 93.76      | 0.09       | 63.50       | 0.51        |
> |            |              | HRank [7]           | ✅          |            | 93.26        | 93.17      | -0.09      | 62.72       | 0.49        |
> |            |              | SANP [4]       | ✅          |            | 93.49        | 93.81      | _0.32_   | 60.24       | -           |
> |            |              | LPSR [17]          |            | ✅          | 93.21        | 93.40      | 0.19       | 60.10       | _0.47_      |
> |            |              | SSL [13]          |            | ✅          | 93.37        | 93.25      | -0.12      | 59.79       | 0.50        |
> |            |              | ELC [14]          |            | ✅          | 93.45        | 93.66      | 0.21       | _58.30_     | -           |
> |            |              | AMSC (Ours)                    | ✅          | ✅          | 93.37        | 93.71      | **0.34**     | **51.91**   | **0.47**    |
> |            | ResNet110    | DBP [12]              |            | ✅          | 93.97        | 93.61      | -0.36      | 141.90      | -           |
> |            |              | GAL [8]           | ✅          | ✅          | 93.50        | 92.55      | -0.95      | 130.20      | 0.95        |
> |            |              | DLP [6] |            | ✅          | -            | -          | _-0.25_    | 129.70      | 1.02        |
> |            |              | ELC [14]          |            | ✅          | 93.60        | 94.07      | **0.47**   | 92.30       | -           |
> |            |              | HRank [7]           | ✅          |            | 93.50        | 92.65      | -0.85      | 79.30       | 0.70        |
> |            |              | DECORE [1]      | ✅          |            | 93.50        | 92.71      | -0.79      | 58.16       | _0.35_      |
> |            |              | AMSC (Ours)                    | ✅          | ✅          | 93.51        | 92.73      | -0.78      | **54.71**   | **0.34**    |

---

> ### Author Response · Authors · 2024-11-25
> **Part2: The response to weaknesses 2.**
>
> **W2:** The authors should report the results of VGG-16 on CIFAR-10 and ResNet-50 on ImageNet, as numerous methods conduct experiments on these architectures. Reporting these results would facilitate comparisons with other approaches.
>
> **A2:** Following your suggestions, we further compare the proposed AMSC with several state-of-the-art (SOTA) pruning methods for VGG16 on CIFAR-10 and ResNet50 on ImageNet. As shown in Table 2, **AMSC achieves the fewer FLOPs with a competitive accuracy and parameter counts for both VGG16 and ResNet50.**
>
> **Table2: Performance comparisons for various architectures on CIFAR-10 and ImageNet.** Pruned and Acc$\uparrow$ denote pruned accuracy and relative accuracy increase, respectively. W and D indicate whether the model is pruned along depth and width. The highest scores in each block are highlighted in bold, while the second highest scores are italics.
>
> | Dataset    | Architecture | Methods                  | W   | D   | Baseline (%) | Pruned (%) | Acc$\uparrow$(%)      | FLOPs (M/B)   | Params. (M)  |
> |:----------:|:------------:|:------------------------:|:---:|:---:|:------------:|:----------:|:--------------:|:-------------:|:------------:|
> | CIFAR-10   | VGG16        | SDN [2] |     | ✅   | 93.50        | 93.47      | -0.03          | 191.38        | **1.79**     |
> |            |              | GAL [8]     | ✅   | ✅   | 93.96        | 92.03      | -1.93          | 189.49        | 3.36         |
> |            |              | HRank [7]     | ✅   |     | 93.96        | 93.43      | -0.53          | 145.61        | 2.51         |
> |            |              | ELC [14]    |     | ✅   | 93.56        | 93.25      | -0.31          | 144.90        | -            |
> |            |              | APIB [5]  | ✅   |     | 93.68        | 94.08      | **0.40**       | 125.30        | 3.55         |
> |            |              | CPMC [15]    | ✅   |     | 93.68        | 93.40      | -0.28          | 106.50        | -            |
> |            |              | AMSC (Ours)             | ✅   | ✅   | 93.60        | 93.46      | _-0.14_        | **100.98**    | _1.96_       |
> | ImageNet   | ResNet50     | GAL [8]     | ✅   | ✅   | 76.15        | 71.95      | -4.20          | 2.33          | 21.20        |
> |            |              | HRank [7]     | ✅   |     | 76.15        | 74.98      | -1.17          | 2.30          | _16.15_      |
> |            |              | AKECP [16]   | ✅   |     | 76.52        | 76.20      | _-0.32_        | 2.29          | **15.16**    |
> |            |              | Greg-2 [10]  | ✅   |     | 76.13        | 75.36      | -0.77          | **1.77**      | -            |
> |            |              | GFP [9]       | ✅   |     | 76.79        | 76.42      | -0.37          | 2.04          | -            |
> |            |              | DepGraph [3] | ✅   | ✅   | 76.15        | 75.83      | **-0.32**      | 1.99          | -            |
> |            |              | AMSC (Ours)             | ✅   | ✅   | 76.15        | 75.53      | -0.62          | _1.85_        | 16.84        |

---

> ### Author Response · Authors · 2024-11-25
> **Part3: The response to weaknesses 3.1 and 3.2.**
>
> **W3:** To demonstrate the effectiveness of Equation 4, more experiments should be conducted in Section 5.3.3, including:
>
> **W31:** Keep $\lambda_0$, $\lambda_1$ unchanged, compare the results of proposed method with those obtained when setting $\lambda(l)=1$, $\lambda(l,j)=1$.
>
>
> **A31:** Since the proposed weight $\sqrt{q _l}/‖\hat{\theta}‖ _2$ ($\sqrt{q _{l,j}}/‖\hat{\theta} _{l,j} ‖ _2$) in AMSC modifies the overall magnitude of the penalty term $\lambda_0 Q_d(\theta)+\lambda_1Q_w(\theta)$, directly comparing AMSC with SSL ($\lambda(l)=1$ and $\lambda(l,j)=1$) under same $\lambda_0$ and $\lambda_1$ leads to an unfair evaluation. Therefore, we have compared the proposed AMSC with SSL using the most suitable parameters for each method in Table 2 and Figure 3 in  the revised manuscript.
>
> As shown in Table 2 in the revised version, AMSC surpasses SSL in accuracy and all compression metrics for ResNet56 on CIFAR-10. To see more clearly, we show the compressed networks in Figure 3. Obviously, SSL treats layers with varying parameter counts equally, leading to a tendency to compress layers with fewer parameters. However, earlier layers typically have fewer parameters, serve as fundamental components for later layers and should be retained. Therefore, pruning those layers will lead to inadequate compression and suboptimal performance. In contrast, AMSC adaptively adjusts the penalty for each layer based on its importance and parameter counts, preserving the earlier layers while compressing the middle layers more extensively. This aligns with the current understanding of neural networks. Particularly, it is well known that the earlier layers usually extract features such as edges, texture and color, which serve as fundamental components for later layers and should be preserved. Conversely, the outputs of the middle layers often show similar features and should be compressed.
>
> **W32:** Compare the results with those obtained using reverse weighting, i.e., when $\frac{\sqrt{q _l}}{‖\hat{\theta} _l ‖ _2}$ is larger, the corresponding weight $\lambda(l)$ should be smaller. Similarly, when $\frac{\sqrt{q _{l,j}}}{‖\hat{\theta} _{l,j} ‖ _2}$ is larger, the corresponding weight $\lambda(l,j)$ should be smaller.
>
> **A32:** We compare the proposed AMSC with the its reverse weighting version in Table 3. Obviously, AMSC achieves more sufficient compression in terms of FLOPs (51.91M v.s. 59.13M) and parameter counts (0.47M v.s. 0.65M), with higher accuracy (93.67\% v.s. 92.07\%). These results align with the original intention behind the weight design in AMSC. Specifically, in our design, $\sqrt{q _l}$ and $\sqrt{q _{l,j}}$ serve as  adaptive tuning parameters for shrinking $\theta _l$ and $\theta _{l,j}$ toward zero based on their respective parameter counts. Meanwhile, $1/‖ \hat{\theta} _l ‖ _2$ and $1/‖ \hat{\theta} _{l,j} ‖ _2$ assign more weights to layers and filters with lower norm values, guiding AMSC to aggressively compress the less important components. **However, reversing these weights causes the method to prioritize compressing components with fewer parameters and lower importance, which ultimately leads to insufficient and inappropriate compression.**
>
> **Table3: Performance comparisons of AMSC with its Reversing Weight (RW) version.**
>
> | Methods   | Accuracy (%) | FLOPs (M)    | Params. (M)   |
> |:---------:|:------------:|:------------:|:-------------:|
> | Baseline  | 93.37        | 125.48       | 0.85          |
> | RW        | 92.07        | 59.13        | 0.65          |
> | AMSC      | **93.71**    | **51.91**    | **0.47**      |

---

> ### Author Response · Authors · 2024-11-25
> **Part4: The response to weaknesses 3.3, 4 and minor issues.**
>
> **W33:** Analyze the influence of $\lambda(l)$ and $\lambda(l,j)$ on accuracy individually, as well as their influence when investigating $\lambda(l)$ and $\lambda(l,j)$ simultaneously.
>
> **A33:** Following your suggestions, we further investigate the individual contributions of $\lambda(l)$ and $\lambda(l,j)$ to accuracy, corresponding to AMSC with $\lambda_1=0$ (depth-only)  and $\lambda_0=0$ (width-only) regularizations, respectively.  We conduct experiments for ResNet56 on CIFAR-10 and VGG16 on CIFAR-100.  As shown in Table 4, **AMSC achieves more effective compression with performance guarantees, outperforming methods that focus solely on width or depth.**
>
> To offer a more intuitive comparison, Figure 4 in the revised manuscript shows how accuracy varies with FLOPs (parameter counts) across the three methods. Notably, under similar FLOPs, AMSC consistently achieves higher accuracy across all architectures and datasets. This demonstrates that multi-dimensional compression is more effective at identifying reasonable  substructures than single-dimensional compression within a given computational budget.
>
> **Table4: Performance comparisons of individual contributions of depth and width regularization on CIFAR-10/100**
>
> | Dataset    | Architecture | Methods  | Accuracy (%) | FLOPs (M)      | Params. (M)   |
> |:----------:|:------------:|:--------:|:------------:|:--------------:|:-------------:|
> | CIFAR-10   | ResNet56     | Baseline | 93.37        | 125.48         | 0.85          |
> |            |              | Width-only   | 93.27        | 86.10          | 0.72          |
> |            |              | Depth-only   | 92.97        | 54.41          | 0.55          |
> |            |              | AMSC     | **93.71**    | **51.91**      | **0.47**      |
> | CIFAR-100  | VGG16        | Baseline | 73.61        | 313.24         | 14.77         |
> |            |              | Width-only   | 72.02        | **127.06**     | **2.00**      |
> |            |              | Depth-only   | 72.82        | 129.03         | 2.32          |
> |            |              | AMSC     | **73.54**    | 148.20         | 2.58          |
>
> **W4:** Do the authors validate the effectiveness of the proposed method on other tasks, such as object detection?
>
> **A4:** Traditional compression literature primarily focuses on classification problems, and we align with these benchmarks in our evaluation. However, our approach is highly scalable and can be easily generalized to other tasks. Specifically, AMSC can be applied to new tasks by simply incorporating the proposed penalty into the objective function of the target task.
>
> **Minor issues:** Several symbol errors. For example, $‖ \theta _l‖ _2 \propto \sqrt{q _l}$ and $‖ \theta _{l,j}‖ _2 \propto \sqrt{q _{l,j}}$ seem to lack rigor. Also, there is an error in the equation on page 5, Line 209.
>
> **A:** Thanks for your reminding, the expression of "$‖ \theta _l‖ _2 \propto \sqrt{q _l}$ and $‖ \theta _{l,j}‖ _2 \propto \sqrt{q _{l,j}}$"  has been changed  to "under the commonly used assumption that $‖ \theta‖ _{\infty} < B$ for some $0< B < +\infty$ in neural network literature [18, 19], $‖ \theta _l‖ _2 \leq \sqrt{q _l}B$ and $‖ \theta _{l,j}‖ _2 \leq  \sqrt{q _{l,j}}B$" in the line 187 of the revised version.
>
> We also correct the error in the equation in the line 212 of the revised version.

---

> ### Author Response · Authors · 2024-11-25
> **Reference**
>
> **Reference**
>
> [1] Manoj Alwani, Yang Wang, and Vashisht Madhavan. Decore: Deep compression with reinforcement learning. In 2022 IEEE/CVF Conference on Computer Vision and Pattern Recognition (CVPR), pages 12339–12349. IEEE, 2022.
>
> [2] Shi Chen and Qi Zhao. Shallowing deep networks: Layer-wise pruning based on feature representations. IEEE Transactions on Pattern Analysis & Machine Intelligence, 41(12):3048–3056, 2019.
>
> [3] Gongfan Fang, Xinyin Ma, Mingli Song, Michael Bi Mi, and Xinchao Wang. Depgraph: Towards any structural pruning. In 2023 IEEE/CVF Conference on Computer Vision and Pattern Recognition (CVPR), pages 16091–16101. IEEE, 2023.
>
> [4] Shangqian Gao, Zeyu Zhang, Yanfu Zhang, Feihu Huang, and Heng Huang. Structural alignment for network pruning through partial regularization. In 2023 IEEE/CVF International Conference on Computer Vision (ICCV), pages 17356–17366. IEEE, 2023.
>
> [5] Song Guo, Lei Zhang, Xiawu Zheng, Yan Wang, Yuchao Li, Fei Chao, Chenglin Wu, Shengchuan Zhang, and Rongrong Ji. Automatic network pruning via hilbert-schmidt independence criterion lasso under information bottleneck principle. In 2023 IEEE/CVF International Conference on Computer Vision (ICCV), pages 17412–17423. IEEE Computer Society, 2023.
>
> [6] Artur Jordao, Maiko Lie, and William Robson Schwartz. Discriminative layer pruning for convolutional neural networks. IEEE Journal of Selected Topics in Signal Processing, 14(4):828–837, 2020.
>
> [7] Mingbao Lin, Rongrong Ji, Yan Wang, Yichen Zhang, Baochang Zhang, Yonghong Tian, and Ling Shao. Hrank: Filter pruning using high-rank feature map. In 2020 IEEE/CVF Conference on Computer Vision and Pattern Recognition (CVPR), pages 1526–1535. IEEE Computer Society, 2020.
>
> [8] Shaohui Lin, Rongrong Ji, Chenqian Yan, Baochang Zhang, Liujuan Cao, Qixiang Ye, Feiyue Huang, and David Doermann. Towards optimal structured cnn pruning via generative adversarial learning. In 2019 IEEE/CVF Conference on Computer Vision and Pattern Recognition (CVPR), pages 2785–2794. IEEE Computer Society, 2019.
>
> [9] L Liu, S Zhang, Z Kuang, A Zhou, J Xue, X Wang, Y Chen, W Yang, Q Liao, and W Zhang. Group fisher pruning for practical network compression. In Proceedings of the 38th International Conference on Machine Learning, volume 139, pages 7021–7032. PMLR: Proceedings of Machine Learning Research, 2021.
>
> [10] Huan Wang, Can Qin, Yulun Zhang, and Yun Fu. Neural pruning via growing regularization. In International Conference on Learning Representations (ICLR), 2021.
>
> [11] Wenxiao Wang, Minghao Chen, Shuai Zhao, Long Chen, Jinming Hu, Haifeng Liu, Deng Cai, Xiaofei He, and Wei Liu. Accelerate cnns from three dimensions: A comprehensive pruning framework. In International Conference on Machine Learning, pages 10717–10726. PMLR, 2021.
>
> [12] Wenxiao Wang, Shuai Zhao, Minghao Chen, Jinming Hu, Deng Cai, and Haifeng Liu. Dbp: Discrimination based block-level pruning for deep model acceleration. arXiv preprint arXiv:1912.10178, 2019.
>
> [13] Wei Wen, Chunpeng Wu, Yandan Wang, Yiran Chen, and Hai Li. Learning structured sparsity in deep neural networks. Advances in Neural Information Processing Systems, 29, 2016.
>
> [14] Jie Wu, Dingshun Zhu, Leyuan Fang, Yue Deng, and Zhun Zhong. Efficient layer compression without pruning. IEEE Transactions on Image Processing, 2023.
>
> [15] Yangchun Yan, Rongzuo Guo, Chao Li, Kang Yang, and Yongjun Xu. Channel pruning via multi-criteria based on weight dependency. In 2021 International Joint Conference on Neural Networks (IJCNN), pages 1–8. IEEE, 2021.
>
> [16] Haonan Zhang, Longjun Liu, Hengyi Zhou, Wenxuan Hou, Hongbin Sun, and Nanning Zheng. Akecp: Adaptive knowledge extraction from feature maps for fast and efficient channel pruning. In Proceedings of the 29th ACM International Conference on Multimedia, pages 648–657, 2021.
>
> [17] Ke Zhang and Guangzhe Liu. Layer pruning for obtaining shallower resnets. IEEE Signal Processing Letters, 29:1172–1176, 2022.
>
> [18] Minshuo Chen, Wenjing Liao, Hongyuan Zha, and Tuo Zhao. Distribution approximation and statistical estimation guarantees of generative adversarial networks. arXiv preprint arXiv:2002.03938, 2020a.
>
> [19] Yuling Jiao, Guohao Shen, Yuanyuan Lin, and Jian Huang. Deep nonparametric regression on approximate manifolds: Nonasymptotic error bounds with polynomial prefactors. The Annals of Statistics, 51(2): 691–716, 2023.

---

> ### Comment · Reviewer_wizb · 2024-11-30
>
> I appreciate the authors for their thoughtful and detailed responses. Most of my concerns have been addressed.
>
> In A1, the proposed method is re-implemented on a new baseline for evaluation. However, the baseline of DenseNet on CIFAR-10 still appears to be lower.
>
> In A2, experiments with VGG-16 on CIFAR-10 and ResNet-50 on ImageNet are reported.
>
> In A3, the effectiveness of Equation 4 is further clarified via additional ablation studies.
>
> In conclusion, while the DenseNet40 baseline on CIFAR-10 remains significantly lower than other methods presented in Table 1, I will increase the rating by one point, as most of my main concerns have been addressed.

---

> > ### Author Response · Authors · 2024-12-02
> > **Thanks for your recognition and valuable feedback**
> >
> > We sincerely appreciate your recognition and valuable feedback.
> > We re-implement the DenseNet40 in a new experimental setting (PyTorch 2.4.0 with an NVDIA GeForce RTX 4090 GPU), and we find that the new baseline could achieve results similar to those reported in other papers. We also have re-implemented the proposed AMSC method on the new baseline, which is entirely based on the code provided in the supplementary material. As shown in the table below, similar conclusion with the previous version can be obtained. We will update this result in the fnal version of the manuscript.
> >
> > **Table: Performance comparisons for DenseNet40 on CIFAR-10.** Pruned and Acc.$\uparrow$ denote pruned accuracy and relative accuracy increase, respectively. W and D indicate whether the model will be pruned along depth and width, respectively. The highest scores in each block are highlighted in bold, while the second highest scores are italics.
> > | Dataset | Architecture | Methods   | W | D | Baseline(\%) | Pruned(\%) | Acc$\uparrow$(\%) | FLOPs(M) | Params.(M) |
> > |:---------:|:------------:|:---------:|:-:|:-:|:-----------:|:---------:|:-------:|:--------:|:----------:|
> > | CIFAR-10 | DenseNet40   | DBP[5]       |   | ✅ | 94.59       | 94.02     | _-0.57_    | 159.25   | 0.43       |
> > |  |    | DECORE[1]    | ✅ |   | 94.81       | 94.04     | -0.77    | 128.13   | _0.37_       |
> > |  |    | GAL[4]       | ✅ | ✅ | 94.81       | 93.53     | -1.28    | 128.11   | 0.45       |
> > |  |    | DHP[2]       | ✅ |   | 94.74       | 93.94     | -0.80    | 112.06   | 0.68       |
> > |  |    | HRank[3]     | ✅ |   | 94.81       | 93.68     | -1.13    | _110.15_   | 0.48       |
> > |  |    | AMSC(Ours)| ✅ | ✅ | 94.65       | 94.14     | **-0.51** | **103.43**| **0.33**   |
> >
> > **Reference**
> >
> > [1] Manoj Alwani, Yang Wang, and Vashisht Madhavan. Decore: Deep compression with reinforcement learning. In 2022
> > IEEE/CVF Conference on Computer Vision and Pattern Recognition (CVPR), pages 12339–12349. IEEE, 2022.
> >
> > [2] Yawei Li, Shuhang Gu, Kai Zhang, Luc Van Gool, and Radu Timofte. Dhp: Differentiable meta pruning via hypernetworks.
> > In Computer Vision–ECCV 2020: 16th European Conference, Glasgow, UK, August 23–28, 2020, Proceedings, Part VIII
> > 16, pages 608–624. Springer, 2020.
> >
> > [3] Mingbao Lin, Rongrong Ji, Yan Wang, Yichen Zhang, Baochang Zhang, Yonghong Tian, and Ling Shao. Hrank: Filter
> > pruning using high-rank feature map. In 2020 IEEE/CVF Conference on Computer Vision and Pattern Recognition
> > (CVPR), pages 1526–1535. IEEE Computer Society, 2020.
> >
> > [4] Shaohui Lin, Rongrong Ji, Chenqian Yan, Baochang Zhang, Liujuan Cao, Qixiang Ye, Feiyue Huang, and David Doermann.
> > Towards optimal structured cnn pruning via generative adversarial learning. In 2019 IEEE/CVF Conference on Computer
> > Vision and Pattern Recognition (CVPR), pages 2785–2794. IEEE Computer Society, 2019.
> >
> > [5] Wenxiao Wang, Shuai Zhao, Minghao Chen, Jinming Hu, Deng Cai, and Haifeng Liu. Dbp: Discrimination based
> > block-level pruning for deep model acceleration. arXiv preprint arXiv:1912.10178, 2019.

---

### Official Review · Reviewer_SQBJ · 2024-11-03

**Soundness:** 3
**Presentation:** 3
**Contribution:** 2
**Rating:** 6
**Confidence:** 3

**Summary:**

This work studies the problem of jointly pruning a neural network along the width (filters in CNNs), and the depth (layers or blocks). The authors propose a multi-dimensional pruning algorithm (AMSC) based on a novel regularization based loss function to identify which elements of the network are to be discarded while training. The elements are then discarded after training. Post pruning, the network is fine-tuned again. The authors suggest novelty of the proposed regulatization term lies in how the authors penalize each filter and layer. The authors provide a theoretical result to show that under certain assumptions, an algorithm like AMSC could potentially obtain optimal width and depth.

**Strengths:**

S1. The problem setting is quite interesting and important.

S2. Writing was mostly easy to follow

S3. Empirical performance of AMSC is better than the baselines

S4. Under the validity of the assumptions, Theorem 1 is quite interesting.

**Weaknesses:**

W1. Writing often lacks precision. Please state terminology before using. Citing a few instances below
- Line 182 "$\hat{\theta}_l$ and $\hat{\theta}_{l,j}$ are estimators.." This statement is unclear
- Line 186. "DNNs are highly unidentifiable" - what does this mean?
- Lines 202 - 213. Writing needs to be improved. Shouldn't $dep$ be a function of $w$ and $wid$ be a function of $l$ since those are constraints?
- Line 239: What is pseudo dimension?

W2. Arguments for Assumption 4.1 and Contribution 1 are not quite convincing for the following reasons
- For figures 2 and 5, why is it that accuracy is pegged to depth / width of the network? That seems unconvincing. Would this hold for different random initialisations of the network?
- Lines 355 - 357: "In contrast..." This argument isn't convincing to ensure validity of Assumption 4.1. Had that been the case, this would have been a lemma/theorem. Figure 1 should be plot for various values of (depth, width). For instance, Fig. 1a is has a fixed width, I suppose. Can this be plot for various widths? Also, how does this plot vary for different random initializations?

W3. Please refer to the questions


Typos:
Line 83-84: The sentence "This phenomenon emphasizes..." seems incomplete

Line 218: Demonstrate~s~

Line 224 & 763: $M_b < \infty$?

Table 1: Tayl~a~(o)r

Line 315: shown

Suggestions:
-  For each mathematical statement (Assumptions, Theorems, Definitions), it would be great if the authors could explain via text or diagram what the statement means.
- Please link Algorithm 1 to the main manuscript.

**Questions:**

Q1. Can the authors contrast or position this work to the body of literature in Neural Architecture Search as a part of related works?

Q2. Why do the authors call AMSC a depth-first strategy? There is no indication in algorithm 1 of this.

Q3. How close are the obtained depth and widths upon using AMSC to the optimal depths and widths for each network?

---

> ### Author Response · Authors · 2024-11-25
> **Part1: The response to weaknesses 1.**
>
> Thank you for your insightful comments and suggestions, which have greatly improved the manuscript.
>
> **W1:** Writing often lacks precision. Please state terminology before using. Citing a few instances below
>
> Line 182 "$\hat{\theta} _l$ and $\hat{\theta} _{l,j}$ are estimators.." This statement is unclear.
>
> Line 186. "DNNs are highly unidentifiable" - what does this mean?
>
> Lines 202 - 213. Writing needs to be improved. Shouldn't $dep$ be a function of $w$ and $wid$ be a function of $l$ since those are constraints?
>
> Line 239: What is pseudo dimension?
>
> **A1:** In the revised manuscript, we have tried our best to clarify statements, including those you mentioned,  and state terminology or provide relevant literature when using specific expression or terminology.  Particularly,
>
> For line 182,  we have changed it to "$\hat{\theta} _l$ and $\hat{\theta} _{l,j}$ are estimators for $\theta _l$ and $\theta _{l,j}$, which can be obtained from the **pre-trained** model", which can be seen in line 186 of the revised version.
>
> For line 186, DNNs are highly unidentifiable, meaning that multiple combinations of depth and width can yield similar performance. We provide relevant literature in  line 190 of the revised version.
>
> For lines 202–213, since DNNs are highly unidentifiable, there may be multiple optimal depth and width values, making their simultaneous determination often impractical.  However, it is feasible to define the optimal depth for a given width and vice versa. Therefore, optimal $dep$ is a function of $w$ and optimal $wid$ is a function of $l$. We describe these definitions in details in lines 205-211 of the revised version.
>
> For line 239, in the statistical learning theory, the pseudo dimension of a function class H is the cardinality of the largest set pseudo-shattered by H [2]. This concept is intrinsically linked to the VC dimension. We provide relevant literature in the line 240 of the revised version.

---

> ### Author Response · Authors · 2024-11-25
> **Part2: The response to weaknesses 2.**
>
> **W2:** Arguments for Assumption 4.1 and Contribution 1 are not quite convincing for the following reasons.
>
> **W21:** For figures 2 and 5, why is it that accuracy is pegged to depth / width of the network? That seems unconvincing. Would this hold for different random initialisations of the network?
>
> **A21:** Accuracy is expressed by the generalization error. According to the generalization theory of DNNs [1,2], the generalization error usually includes two parts: approximation error and stochastic error. Approximation error relies on the ability of the DNNs to approximate the target function, while stochastic error arises when we estimate the DNNs based on the observed finite random samples.
>
> As the model size (e.g., depth and width) increases, the approximation error decreases because the representational capacity of the DNNs improve. However, the stochastic error increases since estimating a larger number of parameters becomes more challenging with limited data [1,2]. Consequently, there exists an optimal model size that minimizes the overall generalization error. **Therefore, if the initial model is large, the accuracy first increases and then decreases as the model size is reduced, as shown in the following Table 1, Figure 2, and Figure 7 in the revised manuscript.**
>
> We further validate this trend with different random initializations in Table 1, which shows that this trend still holds.
>
> **Table1: The accuracy varies with λ₀ and different random initialization when λ₁ = 0 (standard deviation is calculated over five random trials).**
>
> | λ₀            | 0.0          | 0.01         | 0.1          | 0.3          | 0.7          | 0.9          |
> |:-------------:|:------------:|:------------:|:------------:|:------------:|:------------:|:------------:|
> | Accuracy      | 93.28 (0.07) | 93.63 (0.10) | 93.35 (0.10) | 93.03 (0.22) | 92.34 (0.09) | 92.06 (0.12) |
>
> **W22:** Lines 355 - 357: "In contrast..." This argument isn't convincing to ensure validity of Assumption 4.1. Had that been the case, this would have been a lemma/theorem. Figure 1 should be plot for various values of (depth, width). For instance, Fig. 1a is has a fixed width, I suppose. Can this be plot for various widths? Also, how does this plot vary for different random initializations?
>
> **A22:** Thanks for your comments. The expression in Lines 355 - 357 might be too strong. In the revised version, we have changed it to "In contrast, $Q _d(\theta)$ and $Q _w(\theta)$ assigns weights to each layers and filters  based on their importance,  tending to compress less important layers and filters. This makes $Q _d(\theta)$ and $Q _w(\theta)$ more likely to satisfy Assumption 4.1".
>
> Following your suggestions, we further validate the relationship between $dep(\theta^*)$ and $Q_d(\theta^*)$ in various widths. Table 2 shows that the monotonicity relationship between $dep(\theta^*)$ and $Q_d(\theta^*)$ still holds in various widths. This relationship is further illustrated in Figure 6 in Appendix B.5 in revised manuscript.
>
> **Table2:** The relationship between $dep(\theta^*)$ and $Q_d(\theta^*)$ in various widths. Width is calculated by the mean width of different blocks, while depth is the number of blocks.
>
> **Width = 42.37**
> | Depth | 25   | 16   | 16   | 12   | 9    |
> |:-----:|:----:|:----:|:----:|:----:|:----:|
> |$Q_d(\theta^*)$ | 7.28 | 4.15 | 2.62 | 1.79 | 1.23 |
>
> **Width = 39.18**
> | Depth | 22   | 13   | 12   | 7    | 7    |
> |:-----:|:----:|:----:|:----:|:----:|:----:|
> | $Q_d(\theta^*)$ | 4.90 | 2.67 | 2.03 | 1.57 | 1.08 |
>
> **Width = 33.71**
> | Depth | 19   | 12   | 10   | 9    | 8    |
> |:-----:|:----:|:----:|:----:|:----:|:----:|
> | $Q_d(\theta^*)$ | 3.90 | 2.36 | 1.41 | 1.32 | 0.78 |
>
>
>
> We further validate this relationship in different random initializations in Table 3, which shows a same trend in different random initializations.
>
> **Table3:** The relationship between $\text{dep}(\theta^*)$ and $Q_d(\theta^*)$ in different random initializations.
>
> **Seed = 2024**
> | Depth        | 27   | 26   | 25   | 16   | 12   | 9    |
> |:------------:|:----:|:----:|:----:|:----:|:----:|:----:|
> | $Q_d(\theta^*)$        | 9.69 | 9.60 | 7.28 | 4.15 | 1.79 | 1.23 |
>
> **Seed = 2025**
> | Depth        | 27   | 26   | 26   | 17   | 10   | 9    |
> |:------------:|:----:|:----:|:----:|:----:|:----:|:----:|
> | $Q_d(\theta^*)$        | 9.71 | 9.58 | 6.97 | 3.61 | 1.76 | 1.48 |
>
> **Seed = 2026**
> | Depth        | 27   | 26   | 24   | 18   | 10   | 7    |
> |:------------:|:----:|:----:|:----:|:----:|:----:|:----:|
> | $Q_d(\theta^*)$        | 9.72 | 9.61 | 6.97 | 4.07 | 1.99 | 1.62 |

---

> ### Author Response · Authors · 2024-11-25
> **Part3: The response to suggestions 1,2 and questions 1,2,3**
>
> Thank you for your careful review. We have done our best to correct the typos, including those you pointed out, in the revised version.
>
> **S1:** For each mathematical statement (Assumptions, Theorems, Definitions), it would be great if the authors could explain via text or diagram what the statement means.
>
> **A_S1:** In the revised manuscript, for each mathematical statement, we have added the necessary textual explanations. See the paragraphs immediately following Assumptions 4.1, 4.2, and Theorem 4.1, respectively.
>
> **S2:** Please link Algorithm 1 to the main manuscript.
>
> **A_S2:** We have also linked the Algorithm 1 to Section 5.1.
>
>
> **Q1:** Can the authors contrast or position this work to the body of literature in Neural Architecture Search as a part of related works?
>
> **A_Q1:** Following your suggestion, we have added relevant  literature on Neural Architecture Search (NAS) in the line 135-140 of the revised manuscript and discussed the similarities and differences between NAS and our approach.
>
> **Q2:** Why do the authors call AMSC a depth-first strategy? There is no indication in algorithm 1 of this.
>
> **A_Q2:** **DNNs are highly unidentifiable, meaning that multiple combinations of depth and width can yield similar performance. This impedes the simultaneous determination of the optimal network width and depth. However, it is feasible to define the optimal depth for a given width and vice versa.** The proposed AMSC adapt depth-first compression strategy. The rationale for prioritizing depth-wise compression is that reducing depth facilitates model optimization and acceleration compared to reducing width. This depth-first compression strategy involves initially identifying the optimal depth based on a initial width, followed by determining the optimal width based on the identified optimal depth. It should be noted that further depth compression is impossible because the identified width is less than the initial width. Consequently, under this strategy, the depth and width identified by AMSC can be considered as the optimal ones.
>
> Since $\lambda_0$ controls depth, and $\lambda_1$ controls width, AMSC implements a depth-first strategy by first selecting $\lambda_0$ without width controlling ($\lambda_1=0$). Then, with $\lambda_0$ fixed at the selected value, $\lambda_1$ is chosen to specify the width. Therefore, the principle of the depth-first strategy in Algorithm 1 is primarily reflected in the sequential selection of $\lambda_0$ and $\lambda_1$. We have added the related description on Section 5.1 and Appendix B.3.
>
> According to our presented Theorem 4.1 in manuscript, as long as the conditions $\lambda_0 = o_p(1)$, $\lambda_1 = o_p(\lambda_0)$, and $S_n = o_p(\lambda_1)$ are satisfied, the network structure obtained through AMSC can achieve the optimal depth and width under depth-first strategy.  Here, $S_n$ is the stochastic error.
>
> **Q3:** How close are the obtained depth and widths upon using AMSC to the optimal depths and widths for each network?
>
> **A_Q3:** As mentioned in A_Q2, DNNs are highly unidentifiable, meaning that multiple combinations of depth and width can yield similar performance, which may result in multiple optimal depth and width values. This makes the definition of optimal depth and width often impractical.
>
> To make the comparison feasible, we fix either the depth or the width and evaluate the optimality of the other dimension. The resulting depth (or width) can then be considered the optimal achievable value for the network in that dimension.
>
> Table 4 presents the results for AMSC, depth-only (D-only), and width-only (W-only), showing that the depth achieved by AMSC is consistent with that of the D-only method, aligning with our depth-first compression strategy. Meanwhile, the width achieved by AMSC is wider than that of the W-only method but narrower than that of the D-only method. This suggests that AMSC identifies a narrower network under the shallowest network configuration. Additionally, the model size (parameter count) and FLOPs achieved by AMSC are the smallest among the three methods.
>
>
> **Table4: The structures obtained by different settings for ResNet56 on CIFAR-10.** Width is measured by the mean width of different blocks, while depth is the number of blocks.
>
> | Methods | Width      | Depth      | FLOPs (M)    | Params. (M)   |
> |:-------:|:----------:|:----------:|:------------:|:-------------:|
> | W-only  | **30.54**  | 24         | 86.10        | 0.72          |
> | D-only  | 42.92      | **13**     | 54.41        | 0.55          |
> | AMSC    | 39.15      | **13**     | **51.91**    | **0.47**      |

---

> ### Author Response · Authors · 2024-11-26
> **Reference**
>
> **Reference**
>
> [1] Jian Huang, Yuling Jiao, Zhen Li, Shiao Liu, Yang Wang, and Yunfei Yang. An error analysis of generative adversarial networks for learning distributions. Journal of Machine Learning Research, 23(116):1–43, 2022.
>
> [2] Yuling Jiao, Guohao Shen, Yuanyuan Lin, and Jian Huang. Deep nonparametric regression on approximate manifolds: Nonasymptotic error bounds with polynomial prefactors. The Annals of Statistics, 51(2):691–716, 2023.

---

> ### Comment · Reviewer_SQBJ · 2024-11-28
> **Thank you for you rebuttal**
>
> I want to thank the authors for their time during the rebuttal. Per the rebuttal on the comments made by all the reviewers, my initial concerns have been addressed. As suggested by Reviewer GyUQ, ResNet-50 on ImageNet experiments are quite important. It's ok to not be SoTA. I enjoy the different viewpoint of pruning, via the theoretical analysis, this work presents. Having said this, I would request the authors to also baseline ResNet-50 on ImageNet against more stronger pruning algorithms, to my best awareness, such as
>
> [a] ResRep: Lossless CNN Pruning via Decoupling Remembering and Forgetting. Ding et al. ICCV 2021
>
> [b] Structural Pruning via Latency-Saliency Knapsack. Shen et al. NeurIPS 2022
>
> [c] DFPC: Data flow driven pruning of coupled channels without data. Narshana et al. ICLR 2023
>
> Having said this, I am happy to increase my score by one point for overall rating, soundness and presentation.

---

> ### Author Response · Authors · 2024-11-29
> **Thanks for your recognition**
>
> We sincerely appreciate your recognition. We have included the suggested baselines in the table below, and this will be updated in the final version of the manuscript.
>
> **Table: Performance comparisons for ResNet50 on ImageNet.** Pruned and Acc.$\uparrow$ denote pruned accuracy and relative accuracy increase, respectively. W and D indicate whether the model will be pruned along depth and width, respectively. The highest scores in each block are highlighted in bold, while the second highest scores are italics.
>
> | Dataset | Architecture      | Methods | W | D | Baseline(%) | Pruned(%) | Acc.$\uparrow$(%) | FLOPs(B) | Params.(M) |
> |:-------:|:-----------------:|:-------:|:-:|:-:|:-----------:|:---------:|:--------:|:-------:|:----------:|
> | ImageNet | ResNet50 | HALP-55\% [7] | ✅ |   | 76.20 | 76.50 | **0.30** | 2.55 | - |
> |  |  | GAL [4] | ✅ | ✅ | 76.15 | 71.95 | -4.20 | 2.33 | 21.20 |
> |  |  | HRank [3] | ✅ |   | 76.15 | 74.98 | -1.17 | 2.30 | _16.15_ |
> |  |  | AKECP [9] | ✅ |   | 76.52 | 76.20 | -0.32 | 2.29 | **15.16** |
> |  |  | GFP [5] | ✅ |   | 76.79 | 76.42 | -0.37 | 2.04 | - |
> |  |  | DFPC(30) [6] | ✅ |   | 76.10 | 75.90 | -0.20 | 2.07 | - |
> |  |  | DepGraph [2] | ✅ | ✅ | 76.15 | 75.83 | -0.32 | 1.99 | - |
> |  |  | ResRep [1] | ✅ |   | 76.15 | 76.15 | _0.00_ | 1.88 | - |
> |  |  | Greg-2 [8] | ✅ |   | 76.13 | 75.36 | -0.77 | **1.77** | - |
> |  |  | AMSC(Ours) | ✅ | ✅ | 76.15 | 75.53 | -0.62 | _1.85_ | 16.84 |
>
>
>
> **Reference**
>
> [1] Xiaohan Ding, Tianxiang Hao, Jianchao Tan, Ji Liu, Jungong Han, Yuchen Guo, and Guiguang Ding. Resrep: Lossless
> cnn pruning via decoupling remembering and forgetting. In 2021 IEEE/CVF International Conference on Computer
> Vision (ICCV), pages 4490–4500. IEEE Computer Society, 2021.
>
> [2] Gongfan Fang, Xinyin Ma, Mingli Song, Michael Bi Mi, and Xinchao Wang. Depgraph: Towards any structural pruning.
> In 2023 IEEE/CVF Conference on Computer Vision and Pattern Recognition (CVPR), pages 16091–16101. IEEE, 2023.
>
> [3] Mingbao Lin, Rongrong Ji, Yan Wang, Yichen Zhang, Baochang Zhang, Yonghong Tian, and Ling Shao. Hrank: Filter
> pruning using high-rank feature map. In 2020 IEEE/CVF Conference on Computer Vision and Pattern Recognition
> (CVPR), pages 1526–1535. IEEE Computer Society, 2020.
>
> [4] Shaohui Lin, Rongrong Ji, Chenqian Yan, Baochang Zhang, Liujuan Cao, Qixiang Ye, Feiyue Huang, and David Doermann.
> Towards optimal structured cnn pruning via generative adversarial learning. In 2019 IEEE/CVF Conference on Computer
> Vision and Pattern Recognition (CVPR), pages 2785–2794. IEEE Computer Society, 2019.
>
> [5] L Liu, S Zhang, Z Kuang, A Zhou, J Xue, X Wang, Y Chen, W Yang, Q Liao, and W Zhang. Group fisher pruning for
> practical network compression. In Proceedings of the 38th International Conference on Machine Learning, volume 139,
> pages 7021–7032. PMLR: Proceedings of Machine Learning Research, 2021.
>
> [6] Tanay Narshana, Chaitanya Murti, and Chiranjib Bhattacharyya. Dfpc: Data flow driven pruning of coupled channels
> without data. In International Conference on Learning Representations (ICLR), 2022.
>
> [7] Maying Shen, Hongxu Yin, Pavlo Molchanov, Lei Mao, Jianna Liu, and Jose M Alvarez. Structural pruning via
> latency-saliency knapsack. Advances in Neural Information Processing Systems, 35:12894–12908, 2022.
>
> [8] Huan Wang, Can Qin, Yulun Zhang, and Yun Fu. Neural pruning via growing regularization. In International Conference
> on Learning Representations (ICLR), 2021.
>
> [9] Haonan Zhang, Longjun Liu, Hengyi Zhou, Wenxuan Hou, Hongbin Sun, and Nanning Zheng. Akecp: Adaptive knowledge
> extraction from feature maps for fast and efficient channel pruning. In Proceedings of the 29th ACM International
> Conference on Multimedia, pages 648–657, 2021.

---

### Official Review · Reviewer_GyUQ · 2024-11-04

**Soundness:** 2
**Presentation:** 3
**Contribution:** 2
**Rating:** 5
**Confidence:** 5

**Summary:**

This paper proposes a novel Adaptive Multi-dimensional Structured Compression (AMSC) approach that simultaneously compresses the depth and width of neural networks while maintaining model performance. The method demonstrates a thoughtful balance between architectural efficiency and computational effectiveness. The key innovation is incorporating layer- and filter-specific information into penalty terms to adaptively identify redundant components, where Eq 3 is at the center of the paper, which introduces loss terms during training to compress the model by constraining both its depth and width. Experiments suggest the merits of the method vs other counterparts.

**Strengths:**

1. The proposed adaptive weighting scheme ($\lambda(l) = \sqrt{q_l}/||\hat\theta_l||_2$) provides a mathematically principled approach to network compression by automatically adjusting compression intensity based on both parameter counts and layer importance, addressing a fundamental limitation in existing methods that use uniform compression.

**Weaknesses:**

* The method's strong dependency on hyperparameters $λ_0$ and $λ_1$ (as shown in Figure 2) raises concerns about its practical deployability. The paper does not provide a systematic strategy for determining these crucial hyperparameters across different architectures. Tuning these hyper-params needs domain knowledge in practice, which makes the method hard to use.

* The baseline performance of the proposed AMSC method is consistently lower than competing methods across different architectures (e.g., 92.86% vs. 93.03%-93.72% on ResNet56, 92.63% vs. 93.50%-93.97% on ResNet110). This discrepancy in baseline accuracy makes it challenging to fairly evaluate the actual effectiveness of the pruning method, as the reported accuracy improvements might be partially attributed to the lower starting point rather than the superiority of the proposed approach. A more rigorous comparison would require experiments with standardized baseline models.

* Lack of ablation studies to validate the individual contributions of depth and width regularization. Comparison between "depth-only", "width-only", and the full method would better justify the necessity of combining both regularization. It is suggested to conduct ablation studies comparing "depth-only", "width-only", and the full method across different architectures and datasets.

* Tab. 2, SSL is a quite old method. I am not sure if the comparison with it can really show any advantage of the method.

* For pruning papers, Resnet50 on ImageNet is a standard benchmark. This comparison is missing. It is recommended to have it.

**Questions:**

NA

---

> ### Author Response · Authors · 2024-11-25
> **Part1: The response to weaknesses 1.**
>
> Thank you for your insightful comments and suggestions, which have greatly improved the manuscript.
>
> **W1:** The method's strong dependency on hyperparameters $\lambda_0$
> and $\lambda_1$ (as shown in Figure 2) raises concerns about its practical deployability. The paper does not provide a systematic strategy for determining these crucial hyperparameters across different architectures. Tuning these hyperparams needs domain knowledge in practice, which makes the method hard to use.
>
> **A1:** Thanks for your comments. We have added a systematic strategy  for determining these hyperparameters in Section 5.1 and Appendix B.3. We select $\lambda_0$ and $\lambda_1$ by minimizing the prediction errors on the validation set. In AMSC, $\lambda_0$ controls depth and $\lambda_1$ controls width. Following our depth-first strategy, these parameters are selected separately. Specifically, $\lambda_0$ is first selected without considering width compression (i.e., with $\lambda_1=0$). Then, keeping  $\lambda_0$ fixed at the selected value, $\lambda_1$ is chosen to specify the width.
>
> The computational cost for selecting $\lambda_0$ and $\lambda_1$ is very limited due to the  separate selection  strategy. Based on resulting models from few hyperparameters, we can quickly identify the nearly optimal $\lambda_0$ and $\lambda_1$. On average, three adjustments yield satisfactory results.

---

> ### Author Response · Authors · 2024-11-25
> **Part2: The response to weaknesses 2.**
>
> **W2:** The baseline performance of the proposed AMSC method is consistently lower than competing methods across different architectures (e.g., 92.86\% vs. 93.03\%-93.72\% on ResNet56, 92.63\% vs. 93.50\%-93.97\% on ResNet110). This discrepancy in baseline accuracy makes it challenging to fairly evaluate the actual effectiveness of the pruning method, as the reported accuracy improvements might be partially attributed to the lower starting point rather than the superiority of the proposed approach. A more rigorous comparison would require experiments with standardized baseline models.
>
>
> **A2:** The baseline implementations for ResNet56 and ResNet110 on CIFAR-10 are sourced directly from the GitHub repository: https://github.com/akamaster/pytorch_resnet_cifar10. The previous experiments are conducted using PyTorch 1.12.1 on an NVIDIA A40 GPU where the reproduced results are slightly lower than  those reported in the repository. However, when tested in a new experimental environment (PyTorch 2.4.0 with an NVIDIA GeForce RTX 4090 GPU), we find that it could achieve results similar to those reported in other papers. **This indicates that the fluctuation of baseline results might caused by the variation of experimental environments.**
>
> We re-implement the proposed AMSC method on a new baseline for evaluation, which is completely based on the code attached in the supplementary. As shown in the following Table 1, the updated baselines for ResNet56 and ResNet110 on CIFAR-10 achieve accuracies of 93.37\% and 93.51\%, respectively. Using AMSC, these models attain accuracies of 93.71\% and 92.73\%, while simultaneously reducing computational cost by 58.63\% and 78.37\% and parameters by 44.70\% and 80.34\%, respectively, establishing new state-of-the-art (SOTA) results for these architectures.
>
> **Table 1: Performance comparisons for various architectures on CIFAR-10.** Pruned and Acc.$\uparrow$ denote pruned accuracy and relative accuracy increase, respectively. W and D indicate whether the model will be pruned along depth and width, respectively.The highest scores in each block are highlighted in bold, while the second highest scores are italics.
> | Dataset    | Architecture | Methods                        | W          | D          | Baseline (%) | Pruned (%) | Acc$\uparrow$(%)  | FLOPs (M/B) | Params. (M) |
> |:----------:|:------------:|:------------------------------:|:----------------:|:----------------:|:---------------------:|:-------------------:|:---------------------:|:-----------------:|:-----------------:|
> | CIFAR-10   | ResNet56     | GAL [8]           | ✅          | ✅          | 93.26        | 93.38      | 0.12       | 78.74       | 0.75        |
> |            |              | DLP [6] |            | ✅          | -            | -          | -0.82      | 65.80       | 0.52        |
> |            |              | TDPF [11]      | ✅          | ✅          | 93.69        | 93.76      | 0.09       | 63.50       | 0.51        |
> |            |              | HRank [7]           | ✅          |            | 93.26        | 93.17      | -0.09      | 62.72       | 0.49        |
> |            |              | SANP [4]       | ✅          |            | 93.49        | 93.81      | _0.32_   | 60.24       | -           |
> |            |              | LPSR [17]          |            | ✅          | 93.21        | 93.40      | 0.19       | 60.10       | _0.47_      |
> |            |              | SSL [13]          |            | ✅          | 93.37        | 93.25      | -0.12      | 59.79       | 0.50        |
> |            |              | ELC [14]          |            | ✅          | 93.45        | 93.66      | 0.21       | _58.30_     | -           |
> |            |              | AMSC (Ours)                    | ✅          | ✅          | 93.37        | 93.71      | **0.34**     | **51.91**   | **0.47**    |
> |            | ResNet110    | DBP [12]              |            | ✅          | 93.97        | 93.61      | -0.36      | 141.90      | -           |
> |            |              | GAL [8]           | ✅          | ✅          | 93.50        | 92.55      | -0.95      | 130.20      | 0.95        |
> |            |              | DLP [6] |            | ✅          | -            | -          | _-0.25_    | 129.70      | 1.02        |
> |            |              | ELC [14]          |            | ✅          | 93.60        | 94.07      | **0.47**   | 92.30       | -           |
> |            |              | HRank [7]           | ✅          |            | 93.50        | 92.65      | -0.85      | 79.30       | 0.70        |
> |            |              | DECORE [1]      | ✅          |            | 93.50        | 92.71      | -0.79      | 58.16       | _0.35_      |
> |            |              | AMSC (Ours)                    | ✅          | ✅          | 93.51        | 92.73      | -0.78      | **54.71**   | **0.34**    |

---

> ### Author Response · Authors · 2024-11-25
> **Part3: The response to weaknesses 3 and 4.**
>
> **W3:** Lack of ablation studies to validate the individual contributions of depth and width regularization. Comparison between "depth-only", "width-only", and the full method would better justify the necessity of combining both regularization. It is suggested to conduct ablation studies comparing "depth-only", "width-only", and the full method across different architectures and datasets.
>
> **A3:** Following your suggestions, we further investigate the individual contributions of depth and width regularization. We conduct experiments for ResNet56 on CIFAR-10 and VGG16 on CIFAR-100. We implement the width-only and depth-only methods by setting the depth penalty to zero ($\lambda_0=0$) and the width penalty to zero ($\lambda_1=0$) in AMSC, respectively.  As shown in Table 2, **AMSC achieves more effective compression with performance guarantees, outperforming methods that focus solely on width or depth.**
>
> To offer a more intuitive comparison, Figure 4 in the revised manuscript shows how accuracy varies with FLOPs (parameter counts) across the three methods. Notably, under similar FLOPs, AMSC consistently achieves higher accuracy across all architectures and datasets. This demonstrates that multi-dimensional compression is more effective at identifying reasonable  substructures than single-dimensional compression within a given computational budget.
>
> **Table2: Performance comparisons of individual contributions of depth and width regularization on CIFAR-10/100.**
>
> | Dataset    | Architecture | Methods  | Accuracy (%) | FLOPs (M)      | Params. (M)   |
> |:----------:|:------------:|:--------:|:------------:|:--------------:|:-------------:|
> | CIFAR-10   | ResNet56     | Baseline | 93.37        | 125.48         | 0.85          |
> |            |              | W-only   | 93.27        | 86.10          | 0.72          |
> |            |              | D-only   | 92.97        | 54.41          | 0.55          |
> |            |              | AMSC     | **93.71**    | **51.91**      | **0.47**      |
> | CIFAR-100  | VGG16        | Baseline | 73.61        | 313.24         | 14.77         |
> |            |              | W-only   | 72.02        | **127.06**     | **2.00**      |
> |            |              | D-only   | 72.82        | 129.03         | 2.32          |
> |            |              | AMSC     | **73.54**    | 148.20         | 2.58          |
>
> **W4:** Tab. 2, SSL is a quite old method. I am not sure if the comparison with it can really show any advantage of the method.
>
> **A4:** **The main purpose of Table 2 in the manuscript is to investigate the impact of the choice of $\lambda(l)$ and $\lambda(l,j)$.** Particularly, we compare three settings: (1) the proposed AMSC ($\lambda(l)=\sqrt{q _l} / ‖\hat{\theta} _l‖ _2 ,  \lambda(l,j)=\sqrt{q _{l,j}} / ‖ \hat{\theta} _{l,j} ‖ _2$) which incorporates the parameter counts and components importance of layer and filter into the penalty; (2) the SSL ($\lambda(l)=1, \ \lambda(l,j)=1$) which penalizes each layer and filter equally; (3) the GL ($\lambda(l)=\sqrt{q _l}, \lambda(l,j)=\sqrt{q _{l,j}}$) which only incorporates the parameter counts.
>
> As shown in Table 2 in the manuscript, AMSC surpasses SSL in accuracy and all compression metrics, while it also outperforms GL in accuracy and FLOPs, although not in parameter counts. To see more clearly, we show the compressed networks in Figure 3. Obviously, SSL treats layers with varying parameter counts equally, leading to a tendency to compress layers with fewer parameters (Top in Figure 3).  Conversely, GL heavily weights layers with larger parameters, leading to significant  compression of these layers (Bottom in Figure 3). Hence, neither setting achieves precise and efficient compression. In contrast, AMSC adaptively adjusts the penalty for each layer based on its importance and parameter counts, preserving the earlier layers while compressing the middle layers more extensively. This aligns with the current understanding of neural networks. Particularly, it is well known that the earlier layers  usually extract features such as edges, texture and color, which serve as fundamental components for later layers and should be preserved. Conversely, the outputs of the middle layers often show similar features and can be compressed. By setting an appropriate $\lambda(l)$, AMSC effectively distinguishes critical and redundant layers, and achieves a more precise and effective compression.

---

> ### Author Response · Authors · 2024-11-25
> **Part4: The response to weaknesses 5.**
>
> **W5:** For pruning papers, Resnet50 on ImageNet is a standard benchmark. This comparison is missing. It is recommended to have it.
>
> **A5:** Following your suggestions, we further compare the proposed AMSC with several state-of-the-art (SOTA) pruning methods for ResNet50 on ImageNet. As shown in Table 3, **AMSC achieves the fewer FLOPs with a competitive accuracy and parameter counts for ResNet50.**
>
> **Table3: Performance comparisons for ResNet50 on ImageNet.** Pruned and Acc$\uparrow$ denote pruned accuracy and relative accuracy increase, respectively. W and D indicate whether the model is pruned along depth and width. The highest scores in each block are highlighted in bold, while the second highest scores are italics.
>
> | Dataset    | Architecture | Methods                  | W   | D   | Baseline (%) | Pruned (%) | Acc$\uparrow$(%)      | FLOPs (M/B)   | Params. (M)  |
> |:----------:|:------------:|:------------------------:|:---:|:---:|:------------:|:----------:|:--------------:|:-------------:|:------------:|
> | ImageNet   | ResNet50     | GAL [8]     | ✅   | ✅   | 76.15        | 71.95      | -4.20          | 2.33          | 21.20        |
> |            |              | HRank [7]     | ✅   |     | 76.15        | 74.98      | -1.17          | 2.30          | _16.15_      |
> |            |              | AKECP [16]   | ✅   |     | 76.52        | 76.20      | _-0.32_        | 2.29          | **15.16**    |
> |            |              | Greg-2 [10]  | ✅   |     | 76.13        | 75.36      | -0.77          | **1.77**      | -            |
> |            |              | GFP [9]       | ✅   |     | 76.79        | 76.42      | -0.37          | 2.04          | -            |
> |            |              | DepGraph [3] | ✅   | ✅   | 76.15        | 75.83      | **-0.32**      | 1.99          | -            |
> |            |              | AMSC (Ours)             | ✅   | ✅   | 76.15        | 75.53      | -0.62          | _1.85_        | 16.84        |

---

> ### Author Response · Authors · 2024-11-25
> **Reference**
>
> **Reference**
>
> [1] Manoj Alwani, Yang Wang, and Vashisht Madhavan. Decore: Deep compression with reinforcement learning. In 2022 IEEE/CVF Conference on Computer Vision and Pattern Recognition (CVPR), pages 12339–12349. IEEE, 2022.
>
> [2] Shi Chen and Qi Zhao. Shallowing deep networks: Layer-wise pruning based on feature representations. IEEE Transactions on Pattern Analysis & Machine Intelligence, 41(12):3048–3056, 2019.
>
> [3] Gongfan Fang, Xinyin Ma, Mingli Song, Michael Bi Mi, and Xinchao Wang. Depgraph: Towards any structural pruning. In 2023 IEEE/CVF Conference on Computer Vision and Pattern Recognition (CVPR), pages 16091–16101. IEEE, 2023.
>
> [4] Shangqian Gao, Zeyu Zhang, Yanfu Zhang, Feihu Huang, and Heng Huang. Structural alignment for network pruning through partial regularization. In 2023 IEEE/CVF International Conference on Computer Vision (ICCV), pages 17356–17366. IEEE, 2023.
>
> [5] Song Guo, Lei Zhang, Xiawu Zheng, Yan Wang, Yuchao Li, Fei Chao, Chenglin Wu, Shengchuan Zhang, and Rongrong Ji. Automatic network pruning via hilbert-schmidt independence criterion lasso under information bottleneck principle. In 2023 IEEE/CVF International Conference on Computer Vision (ICCV), pages 17412–17423. IEEE Computer Society, 2023.
>
> [6] Artur Jordao, Maiko Lie, and William Robson Schwartz. Discriminative layer pruning for convolutional neural networks. IEEE Journal of Selected Topics in Signal Processing, 14(4):828–837, 2020.
>
> [7] Mingbao Lin, Rongrong Ji, Yan Wang, Yichen Zhang, Baochang Zhang, Yonghong Tian, and Ling Shao. Hrank: Filter pruning using high-rank feature map. In 2020 IEEE/CVF Conference on Computer Vision and Pattern Recognition (CVPR), pages 1526–1535. IEEE Computer Society, 2020.
>
> [8] Shaohui Lin, Rongrong Ji, Chenqian Yan, Baochang Zhang, Liujuan Cao, Qixiang Ye, Feiyue Huang, and David Doermann. Towards optimal structured cnn pruning via generative adversarial learning. In 2019 IEEE/CVF Conference on Computer Vision and Pattern Recognition (CVPR), pages 2785–2794. IEEE Computer Society, 2019.
>
> [9] L Liu, S Zhang, Z Kuang, A Zhou, J Xue, X Wang, Y Chen, W Yang, Q Liao, and W Zhang. Group fisher pruning for practical network compression. In Proceedings of the 38th International Conference on Machine Learning, volume 139, pages 7021–7032. PMLR: Proceedings of Machine Learning Research, 2021.
>
> [10] Huan Wang, Can Qin, Yulun Zhang, and Yun Fu. Neural pruning via growing regularization. In International Conference on Learning Representations (ICLR), 2021.
>
> [11] Wenxiao Wang, Minghao Chen, Shuai Zhao, Long Chen, Jinming Hu, Haifeng Liu, Deng Cai, Xiaofei He, and Wei Liu. Accelerate cnns from three dimensions: A comprehensive pruning framework. In International Conference on Machine Learning, pages 10717–10726. PMLR, 2021.
>
> [12] Wenxiao Wang, Shuai Zhao, Minghao Chen, Jinming Hu, Deng Cai, and Haifeng Liu. Dbp: Discrimination based block-level pruning for deep model acceleration. arXiv preprint arXiv:1912.10178, 2019.
>
> [13] Wei Wen, Chunpeng Wu, Yandan Wang, Yiran Chen, and Hai Li. Learning structured sparsity in deep neural networks. Advances in Neural Information Processing Systems, 29, 2016.
>
> [14] Jie Wu, Dingshun Zhu, Leyuan Fang, Yue Deng, and Zhun Zhong. Efficient layer compression without pruning. IEEE Transactions on Image Processing, 2023.
>
> [15] Yangchun Yan, Rongzuo Guo, Chao Li, Kang Yang, and Yongjun Xu. Channel pruning via multi-criteria based on weight dependency. In 2021 International Joint Conference on Neural Networks (IJCNN), pages 1–8. IEEE, 2021.
>
> [16] Haonan Zhang, Longjun Liu, Hengyi Zhou, Wenxuan Hou, Hongbin Sun, and Nanning Zheng. Akecp: Adaptive knowledge extraction from feature maps for fast and efficient channel pruning. In Proceedings of the 29th ACM International Conference on Multimedia, pages 648–657, 2021.
>
> [17] Ke Zhang and Guangzhe Liu. Layer pruning for obtaining shallower resnets. IEEE Signal Processing Letters, 29:1172–1176, 2022.

---

### Meta-Review · Area_Chair_5pFf · 2024-12-22

**Metareview:**

This paper proposes a novel Adaptive Multi-dimensional Structured Compression method that simultaneously learns the minimal depth, the minimal width, and network parameters under the strategy that prioritizes depth compression. The objective (Eq. 3) enhanced by adaptive weighting in Eq. 4 come with some theoretical guarantees on identifying the minimal depth and width in terms of selection consistency, given depth-first compression strategy. While the theoretical results are interesting, the practical performance is not outstanding: the results often comes with a tradeoff in FLOPs and the accuracy drop compared with baseline methods. Even in the rebuttal results, there were no results showing the method outperforms baselines in both FLOPs and accuracy in all settings. In this case, another thing the authors could do is to tune the hyperparameters of the method to provide multiple results that can outperform the best accuracy accuracy when FLOPs match, and outperform the best FLOPs when accuracy match. Without such results, it is unclear whether the method tends to cause some tradeoff in practice.

**Additional Comments On Reviewer Discussion:**

The reviewers raised concerns about the experimental results and these were addressed by the authors in the rebuttal. However, the results are still not strong in my opinion.

---

### Decision · Program_Chairs · 2025-01-22

Reject